# Structural and mechanistic insights into activation of the human RNA ligase RTCB by Archease

Janina Lara Gerber [1], Suria Itzel Morales Guzmán [1], Lorenz Worf [1], Petra Hubbe [1], Jürgen Kopp [1] & Jirka Peschek [1] ✉

RNA ligases of the RTCB-type play an essential role in tRNA splicing, the unfolded protein response and RNA repair. RTCB is the catalytic subunit of the pentameric human tRNA ligase complex. RNA ligation by the tRNA ligase complex requires GTP-dependent activation of RTCB. This active site guanylylation reaction relies on the activation factor Archease. The mechanistic interplay between both proteins has remained unknown. Here, we report a biochemical and structural analysis of the human RTCB-Archease complex in the pre- and post-activation state. Archease reaches into the active site of RTCB and promotes the formation of a covalent RTCB-GMP intermediate through coordination of GTP and metal ions. During the activation reaction, Archease prevents futile RNA substrate binding to RTCB. Moreover, monomer structures of Archease and RTCB reveal additional states within the RNA ligation mechanism. Taken together, we present structural snapshots along the reaction cycle of the human tRNA ligase.

RNA ligases are found in all domains of life. They catalyze the ligation of RNA molecules via formation of phosphodiester bonds during different RNA processing and repair mechanisms. Among the various types of RNA ligases, a distinct functional class is comprised of tRNA ligases. These enzymes are an essential component of the cellular tRNA splicing machinery[1–3]. In eukaryotes and archaea, intron-containing pre-tRNAs undergo splicing via a fully enzyme-catalyzed reaction[1]. The intron is first excised by a tRNA splicing endonuclease (TSEN) and the resulting tRNA exon halves are sealed by a tRNA ligase to form a mature tRNA[4,5]. Notably, eukaryotic tRNA ligases also catalyze the non-conventional, cytoplasmic splicing of the *XBP1* mRNA during the unfolded protein response (UPR) utilizing a similar mechanism. After cleavage by the transmembrane endonuclease/stress sensor IRE1, the intron is removed and the cleaved *XBP1* exons are sealed by the tRNA ligase complex[6–10].

Two separate enzymatic RNA ligation mechanisms have evolved in eukaryotes. Initially discovered in yeast, Trl1-type ligases are tripartite enzymes with a canonical ATP-dependent ligase domain (i.e., adenylyltransferase), which is similar to T4 RNA ligase 1 and is also present in capping enzymes or DNA ligases[11–14]. Later, the discovery of

RNA 2′,3′-cyclic phosphate and 5′-OH ligase (RTCB) revealed a new class of RNA ligases. RTCB-type ligases are present in bacteria, archaea and metazoa displaying a high level of conservation[15–17], but are absent in fungi and plants[1]. Despite their seemingly independent evolution, both RNA ligase types fulfil the same functional role in eukaryotic cells during tRNA splicing and the UPR: They ligate RNA ends carrying a 2′,3′-cyclic phosphate (cP) and a 5′-OH[1,2].

The biochemical mechanism of RTCB-type ligases is distinctly different from canonical adenylyltransferases[15–17]. RTCB-type ligases require two co-factors: guanosine triphosphate (GTP) and divalent metal ions[17–21]. The RTCB-catalyzed ligation reaction occurs in three metal-dependent steps[18,19]. The first step consists of the activation of RTCB by guanylylation of the active site histidine (His428 in human RTCB) via formation of a phosphoramidate (Nε-Pα) bond. Prior to the second step, RTCB hydrolyzes RNA substrates with a 2′,3′-cP end, which are typically present after endonucleolytic cleavage, into a 3′ phosphate. The covalently bound GMP is then transferred to the 3′ phosphate end of the 5′ exon to form an RNA(3′)-P-P-(5′)G intermediate. Lastly, the 5′-OH of the 3′ exon attacks the activated 3′ end to form a 3′−5′ phosphodiester bond with concomitant release of GMP.

[1]Heidelberg University, Biochemistry Center (BZH), Im Neuenheimer Feld 328, Heidelberg, Germany. ✉e-mail: jirka.peschek@bzh.uni-heidelberg.de

The phosphate in the final exon-3′-P-5′-exon junction originates from the former 2′,3′-cP (or 3′ phosphate).

Crystal structures of human RTCB and its homolog from the archaeon *Pyrococcus horikoshii* have revealed a fold that is entirely different from canonical ATP-dependent ligases[18,19,22,23]. The structures show two metal binding sites (A and B) each coordinating one divalent metal cation. While manganese is the canonical metal co-factor for RTCB-catalyzed ligation, a recent structure of human RTCB contains two cobalt ions instead[23]. The identity of both metal sites and their precise role during catalysis remain unclear.

In metazoa, RTCB functions as the catalytically active subunit of the pentameric tRNA ligase complex (tRNA-LC). The tRNA-LC further consists of the DEAD-box helicase DDX1 and three subunits of unknown function: CGI-99, FAM98B and ASHWIN (ASW)[16]. Mapping of the molecular architecture of the complex via crosslinking-mass spectrometry revealed a network of contact sites between RTCB and all other subunits[23]. In addition, some RTCB-type ligases, like *Ph*RtcB and human RTCB, require the aid of Archease, a conserved small protein (16–20 kDa range), for efficient ligation[24,25]. Archease was initially discovered by common genetic organization with RTCB-type RNA ligases in certain prokaryotes[24]. Both proteins display co-evolution ranging from archaea to higher eukaryotes. While the generally accepted role of Archease is the promotion of RTCB guanylylation, the precise activation mechanism of RTCB-type ligases by Archease has remained unclear. Isolated and pre-activated (i.e.,

guanylylated) RTCB stalls after a single round of ligation, but it can resume upon addition of Archease[24,25]. Thus, multiple-turnover ligation by RTCB requires the presence of Archease. Moreover, *P. horikoshii* Archease alters the nucleotide specificity of *Ph*RtcB by allowing the substitution of GTP for ATP, ITP and dGTP[25]. Crystal structures of prokaryotic and human Archease reveal dimeric assemblies due to an N-terminal strand swap. Moreover, the surface of Archease displays predominantly negative charges[25,26]. However, the mode of interaction with RTCB and the structure of the presumed RTCB-Archease activation complex is unknown. In fact, in vitro interaction studies suggest a rather transient interaction of both proteins despite their functional interplay[23,24,27]. Here, we show crystal structures of the RTCB-Archease activation complex at different catalytic steps and provide biochemical insights into the role of Archease on RTCB-mediated RNA ligation.

## Results

### Archease promotes guanylylation of human RTCB

To dissect the Archease-mediated activation mechanism of RTCB, we first reconstituted the RNA ligase reaction in vitro using purified components. To this end, we purified human Archease (isoform 1) and human RTCB from recombinant expression in *E. coli* and insect cells, respectively (Supplementary Fig. 1). We tested the activity of recombinant human RTCB by in vitro ligation of an RNA substrate derived from the unspliced *XBP1* mRNA (*XBP1*u). We used a previously established bifurcated stem-loop structure (Fig. 1a) that consists of the 26-nt

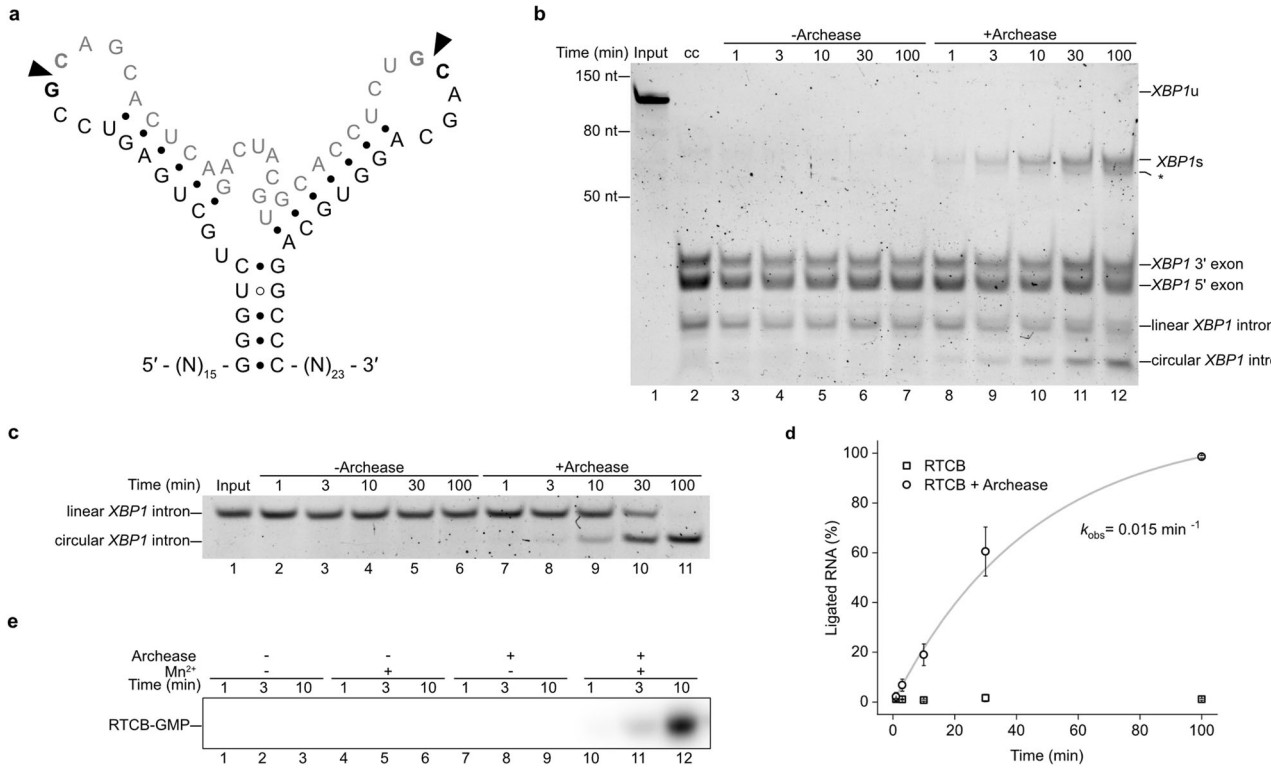

**Fig. 1 | In vitro splicing assay of the RTCB-Archease complex. a** Secondary structure of the bifurcated stem-loop substrate derived from human *XBP1* mRNA. Arrowheads indicate the splice sites, the intronic sequence is depicted in grey. **b** In vitro ligation of the *XBP1* RNA substrate by RTCB (500 nM) in the presence of the co-factors MnCl$_2$ (0.5 mM) and GTP (1 mM) without or with Archease (1 μM). The *XBP1* mRNA (lane 1; Input) was cleaved by the IRE1 RNase (lane 2; cc: cleavage control). Ligation reactions were performed at 30 °C in triplicate (also for (**c**, **e**)). Aliquots were taken at different time points and analyzed by denaturing urea-PAGE. The asterisk denotes an unknown byproduct of the ligation reaction; for instance, a different conformation of the ligated product or re-ligation of the 5′ exon with the intron. **c** In vitro ligation of *XBP1* intron by RTCB (250 nM) in the presence of the co-

factors MnCl$_2$ (0.5 mM) and GTP (1 mM) in the absence or presence of 1 μM Archease (conditions as in (**b**)). **d** Quantification of the RNA ligation assay in (**c**). The relative amounts of ligated product in the presence of RTCB only (squares) or RTCB with Archease (circles) were fitted to a first-order model to determine the indicated apparent rate constant. Values represent mean and standard deviation of technical replicates (*n* = 3). Individual gels were processed in parallel. **e** RTCB (1 μM) guanylylation assay with radioactive α-$^{32}$P-GTP (80 nM) in the absence or presence of Archease (1 μM) and MnCl$_2$ (0.5 mM) were incubated at 30 °C and analyzed by urea-PAGE and visualized by autoradiography. All assays were performed in triplicate. Source data are provided as a Source Data file.

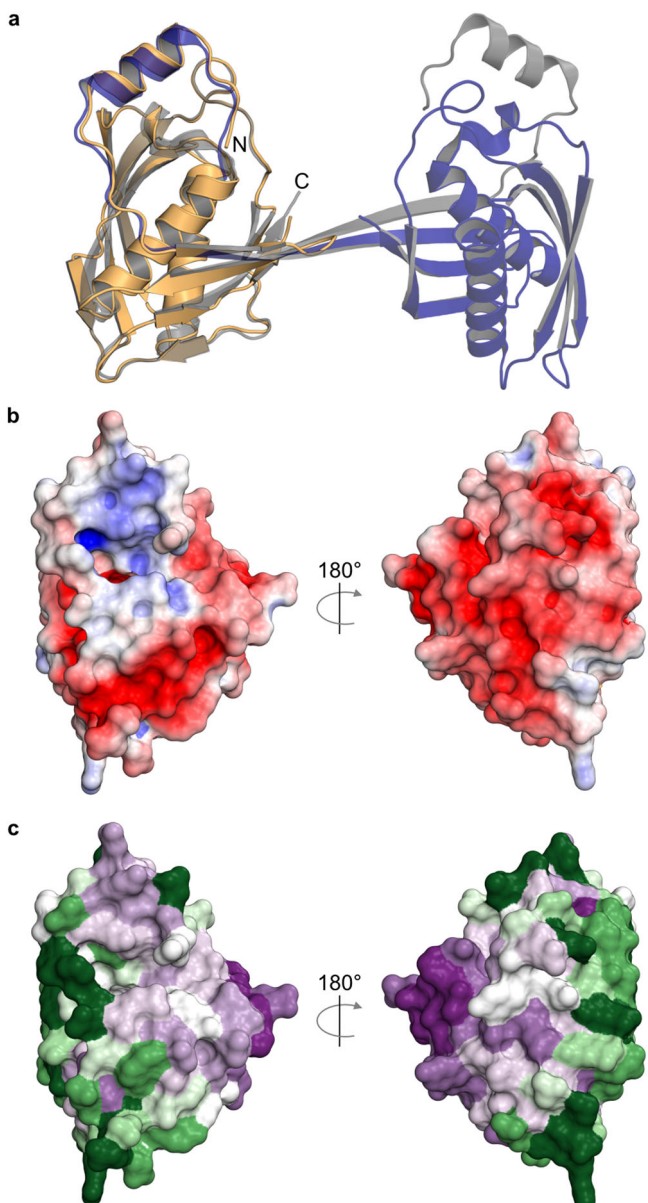

**Fig. 2 | Crystal structure of monomeric human Archease. a** X-ray crystal structure of monomeric human Archease (PDB ID: 8BTX) in orange. The monomer is superimposed with the dimeric human Archease structure (PDB ID: 5YZ1), with the overlaid protomer in purple and the other one in grey. Both structures are depicted in cartoon representation. **b** Electrostatic surface potential map (+/− 5kT/e) of human Archease. Positive potential is represented in blue and negative potential in red. **c** Surface map of the conservation analysis of Archease created with ConSurf out of 500 sequences with a percentual identity between 85% and 25%, using HMMER search algorithm with an *E*-value cut-off of 0.0001. The color represents the degree of conservation from low (dark green) to high (dark purple).

intron RNA with flanking exon segments[10]. After endonucleolytic cleavage by IRE1, we determined the effect of Archease on the RTCB-catalyzed exon-exon ligation in the presence of GTP and $Mn^{2+}$. The production of the spliced RNA fragment (*XBP1s*) was dependent upon addition of Archease to the ligase reaction (Fig. 1b). In the same experiment, we also noted ligation of the intron indicated by a faster migrating circular RNA species (Fig. 1b, lanes 8–12). We used the 26-nt *XBP1* intron sequence harboring a 5′-OH and a 3′ phosphate to yield a single-substrate single-product ligation assay, which confirmed the dependence of human RTCB on Archease (Fig. 1c and Supplementary

Fig. 2a). In the presence of Archease, intron circularization progressed to completion with an apparent rate constant of 0.015 min⁻¹ while we did not detect any ligated product without Archease (Fig. 1d). To confirm the role of Archease during the initial activation of RTCB, which results in the formation of a covalent RTCB-GMP intermediate via phosphoramidate linkage to the active-site histidine, we monitored incorporation of radioactive GTP. We detected the formation of guanylylated RTCB only in the presence of Archease. This activation step was fully dependent on the presence of $Mn^{2+}$ ions (Fig. 1e). Archease did not alter the nucleotide specificity of human RTCB. We did not observe ligation of the *XBP1* intron RNA in the presence of ATP instead of GTP (Supplementary Fig. 2b), which agrees with a previous study using a pre-tRNA substrate[24]. These results confirm the important role of Archease on tRNA ligase activation by promoting the manganese-dependent guanylylation of the enzymatic subunit RTCB.

### Structure of human Archease reveals a compact monomeric fold

Previously reported Archease structures from several organisms show deviating conformations with regards to the N-terminal part. In crystal structures of *P. horikoshii, Thermotoga maritima* and human Archease the N-terminal part (residues 21–55 for *H.s.*) adopts an extended conformation pointing away from the protein[25,26]. The swapping of the N-terminal helix and β-strand with a neighboring Archease molecule creates a dimeric arrangement in the crystal. In contrast, an NMR solution structure of *Methanobacterium thermoautotrophicum* Archease (UniProtAC O27635) shows a globular monomer[28].

To discern between the different conformations and oligomeric states, we crystallized human Archease (isoform 1; UniProtAC A8K0B5) and determined the X-ray structure at 1.84 Å resolution (Fig. 2a). The crystal structure exhibits a globular, arrowhead-shaped monomer, unlike the previously reported structure of the human Archease dimer[26] (Fig. 2a, superposition with PDB ID 5YZ1), with β-strand 1 attached to β-strand 2 by forming a β-hairpin (Supplementary Fig. 3a). The electrostatic surface potential of Archease is predominantly negative with a slightly positive region around the N-terminal helix (Fig. 2b). Residues 22–47 including helix 1 and β-strand 1 are located at positions equivalent to the swapped elements in the dimeric structure of human Archease (Fig. 2a and Supplementary Fig. 3a). Helix 1 and the following loop possess high B-factors suggesting increased flexibility in this conserved region. No metal ion was bound in our Archease structure, unlike a previous structure of the *Ph*Archease dimer (Supplementary Fig. 3b, c), which harbors a calcium ion[25]. Analyzing evolutionary conservation using ConSurf[29,30] revealed a cluster of mostly surface-exposed residues in the conserved tip of the arrowhead-shaped Archease monomer (Fig. 2c). Remarkably, this region was previously suggested to be important for Archease-mediated activation of RTCB[24]. Taken together, our data present a monomeric structure of human Archease without N-terminal strand swapping in contrast to previously crystallized dimeric assemblies.

### Binding of Archease to RTCB is dependent on GTP and $Mn^{2+}$ ions

Despite the established role of Archease as a catalytic activator of RTCB, the physical interaction between both proteins remains poorly characterized. Thus, we tested the interaction between RTCB and Archease using amine-reactive crosslinking in combination with SDS-PAGE analysis (Fig. 3a). Formation of an additional band at about 70 kDa, which corresponds to a one-to-one complex occurred only in the presence of GTP and $Mn^{2+}$ (Fig. 3a, lane 7). There was no detectable formation of the cross-linked species in the absence of either of the two co-factors (Fig. 3a, lanes 4, 5 and 6). We further confirmed this finding by using a fluorescently labeled RTCB (non-specific labeling with AF488) and Archease (site-specific labelling of the S147C variant with AF647). The use of the fluorophore-modified proteins allowed us to confirm the identity of a cross-linked band as the RTCB-Archease

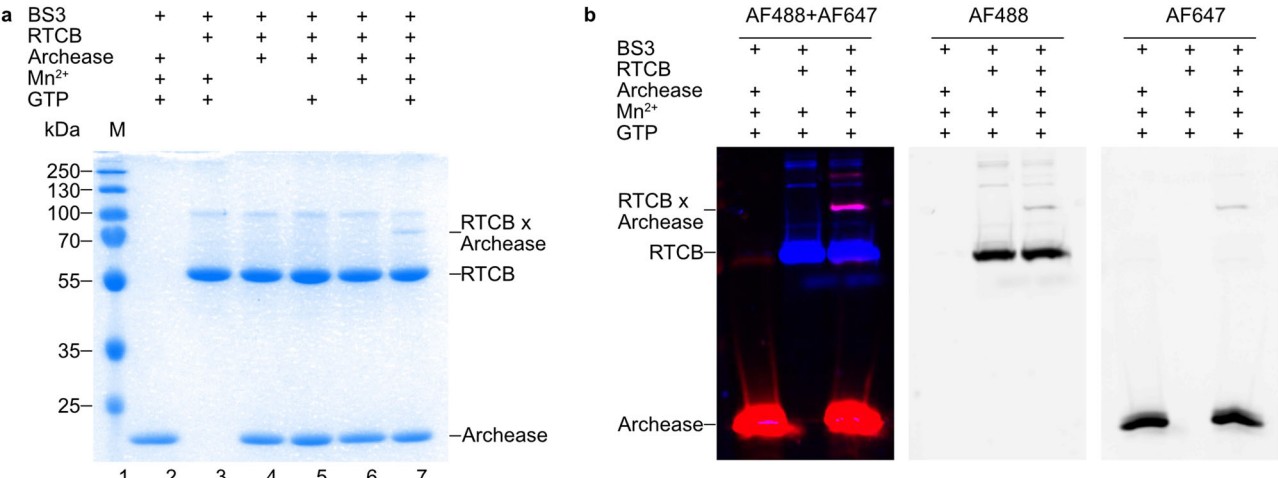

**Fig. 3 | Interaction of human RTCB and Archease is dependent on GTP and Mn²⁺.**
**a** The interaction of RTCB (10 μM) and Archease (10 μM) after crosslinking with bis(sulfosuccinimidyl)suberate (BS3) was analyzed by SDS-PAGE and stained with Coomassie blue. Bands corresponding to the cross-linked RTCB-Archease complex are marked with "RTCB × Archease". All reactions were performed at 30 °C in triplicate (also for (**b**)). **b** The interaction of fluorescently labeled RTCB-AF488 (5 μM) and Archease-S147C-AF647 (5 μM) after crosslinking with BS3 was analyzed by SDS-PAGE and visualized by fluorescence. Bands corresponding to the cross-linked RTCB-Archease complex are marked with "RTCB × Archease". Source data are provided as a Source Data file.

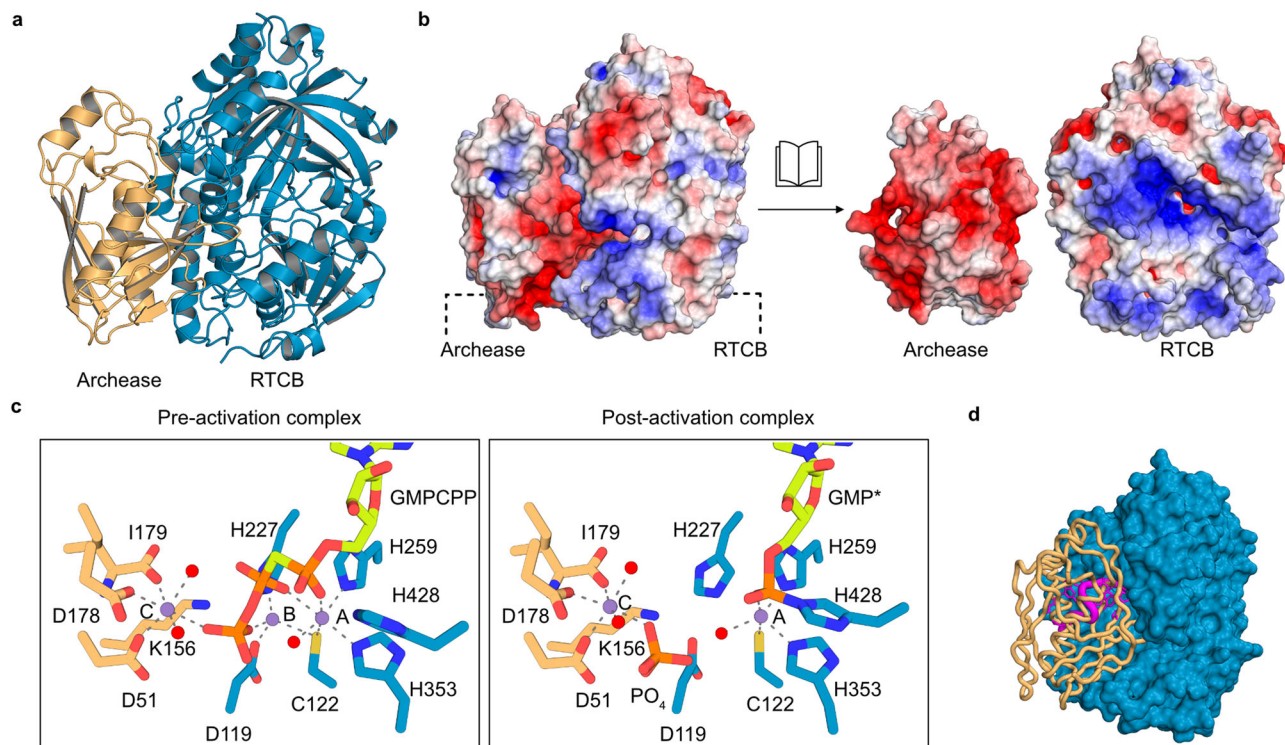

**Fig. 4 | Crystal structure of the of human RTCB-Archease complex. a** X-ray crystal structure of the human RTCB-Archease complex. RTCB (teal) and Archease (orange) are shown in cartoon representation. **b** Electrostatic surface potential map (± 5kT/e) of the RTCB-Archease complex (folded open on the right). Positive potential is represented in blue and negative potential in red. **c** Human RTCB (teal) active site in different activation states in presence of Archease (orange). Mn²⁺ ions are labeled according to their position (A, B, C). Mn²⁺ ions (purple) and waters (red) are presented as spheres. GMP* and GMPCPP are shown as sticks (lime). The hydrogen bonds and metal coordination bonds are shown as grey dashed lines. Left: Pre-activation RTCB active site with GMPCPP as nucleotide co-factor. Right: Post-activation RTCB active site with activator Archease. Mn²⁺-coordinating residues and Archease residue K156 as well as PO₄ are shown as sticks. The guanylylated GMP-H428 is marked as GMP*. **d** Superposition of the human RTCB-Archease complex and *P. horikoshii* RtcB with ssDNA (PDB ID: 7LFQ). The human RTCB (teal) structure is shown in surface view together with human Archease (orange) and the ssDNA (pink), both depicted as ribbons.

complex (Fig. 3b). In both experimental approaches, the majority of the protein remained monomeric, which suggests a transient interaction. Nevertheless, our crosslinking results showed that the RTCB-Archease activation complex forms in the simultaneous presence of GTP and Mn²⁺.

## Crystal structures reveal the architecture of the RTCB-Archease complex

Since crosslinking experiments suggested a stabilization of the complex in the presence of its co-factors, we attempted crystallization of the RTCB-Archease complex with GTP or the α,β-non-hydrolyzable

GTP-analog guanosine-5′-[(α,β)-methyleno]triphosphate (GMPCPP), both in presence of $Mn^{2+}$ ions. We were successful in obtaining diffracting crystals for both combinations in space group $P2_1$ with similar cell dimensions and determined the complex structures at 2.2 Å and 2.3 Å resolution, respectively (Fig. 4a). Using X-ray fluorescence spectroscopy, we confirmed the presence of $Mn^{2+}$ ions in the complex structures while lower amounts of $Zn^{2+}$ ions were also detected (see Source Data). Both structures contain four similar copies of a one-to-one complex between RTCB and Archease in the asymmetric unit. In both complexes the conserved tip of the arrowhead-shaped Archease is reaching directly into the active site of RTCB, thus forming an extensive interface between both proteins with an area of about 2200 Å² (Supplementary Fig. 4a and Supplementary Data 1). The electrostatic surface potential shows the complementarity of a negatively charged Archease surface binding to the predominantly positively charged active site pocket of RTCB (Fig. 4b).

Overall, RTCB and Archease undergo only minor conformational changes upon formation of the activation complex. Within Archease, β-hairpin 46–51 undergoes the largest conformational change compared to the monomeric Archease structure (Supplementary Fig. 4b). Smaller conformational changes occur in the highly conserved region 159–162 and in the N-terminal part, where we could additionally resolve residues 10–21. Unbound and bound Archease superimpose with an overall RMSD ($C_\alpha$) of 0.68 Å. Two additional regions, which are missing in the structure with GMP (PDB ID 7P3B)[23], are resolved in RTCB within our complex structure. These are residues 45–64, which extend a helix and the following loop compared to *Ph*RtcB, and residues 436–444, which form a short helix. Both segments bind directly to Archease and presumably stabilize the complex (Supplementary Fig. 4c)[25].

## Crystal structure of the pre-activation complex

Co-crystallizing with $Mn^{2+}$ and the non-hydrolyzable GTP analog GMPCPP allowed us to obtain an RTCB-Archease complex with a well-resolved GMPCPP molecule in the active site of RTCB. Strikingly, three $Mn^{2+}$ ions were bound in the composite active site formed by RTCB and Archease (Fig. 4c, left). These $Mn^{2+}$ ions occupy three distinct positions A, B and C (hereinafter indicated in parentheses). Positions A and B are exclusively defined by RTCB residues and have been described in previously solved RTCB structures[18,19,23]. Our complex structures revealed an additional metal site, named position C, which solely depends on Archease residues for $Mn^{2+}$ binding. $Mn^{2+}$(A) is octahedrally coordinated by RTCB side chains H259 (*e*; equatorial), H353 (*a*; axial), C122 (*e*), a water molecule (*e*) and two oxygens from the alpha (*e*) and beta (*a*) phosphate moieties of GMPCPP. $Mn^{2+}$(B) is tetrahedrally coordinated by RTCB side chains D119, C122, H227 and an oxygen atom from the γ-phosphate part of GMPCPP. $Mn^{2+}$(C) is coordinated octahedrally by Archease side chains D51 (*a*), D178 (*e*), the terminal carboxyl group of I179 (*e*), an oxygen (*a*) of the γ-phosphate moiety of GMPCPP and two water molecules (*e*). The position of the $Mn^{2+}$ ion in position C matches the localization of the $Ca^{2+}$ ion in the dimeric structure of *Ph*Archease (Supplementary Fig. 4d, e). Base and ribose of the guanosine moiety are bound similarly to other RTCB-type structures (Supplementary Fig. 4f). The α-phosphate oxygens are pointing towards the catalytic H428. Nε2 of H428 is close to the alpha phosphorus atom (3.5 Å). One α-phosphate oxygen is coordinated to $Mn^{2+}$(A). Moreover, H428Nε2, α-phosphorus atom and carbon atom of GMPCPP are nearly in line, i.e., have adopted already the geometry required for the guanylylation reaction. One of the β-phosphates is coordinated to $Mn^{2+}$(A). Two of the three γ-phosphate oxygens are coordinated to $Mn^{2+}$(B) and $Mn^{2+}$(C), respectively. Overall, the structure mimics a pre-activation state in which Archease is already bound but the guanylyl group is not yet transferred to H428 (Fig. 4c, left).

## Crystal structure of the post-activation complex with activated RTCB

To obtain a structure of the RTCB-Archease complex in the post-activation state (i.e., with the activated RTCB-GMP intermediate), we co-crystallized RTCB with Archease in the presence of the nucleotide co-factor GTP, and $Mn^{2+}$ ions. The nucleotide co-factor GTP reacted with the active site histidine H428 to form the guanylylated RTCB-H428-GMP intermediate via a phosphoramidate bond (Fig. 4c, right). The nucleoside part of the covalently bound GMP (indicated as GMP*) adopts a conformation similar to the pre-activation structure with GMPCPP. Our structures confirm the previously identified residues involved in base and sugar binding (Supplementary Fig. 4g)[23]. The position of the covalent bond at the α-phosphate was inverted due to the phosphoryl transfer reaction (compare left to right panel in Fig. 4c and Supplementary Fig. 4h), while the coordination of one of the α-phosphate oxygens to $Mn^{2+}$(A) remained unaffected. With its additional ligands from RTCB side-chains C122, H259 and H353 as well as a water molecule, $Mn^{2+}$(A) shows a square pyramidal coordination. No metal ion was detected at the $Mn^{2+}$(B) site described above. The $Mn^{2+}$(C) site is occupied and coordinated octahedrally as in the GMPCPP structure. We assigned density to a single phosphate group at the axial position, corresponding to the γ-phosphate in the GMPCPP structure. In addition to metal coordination, the phosphate molecule is bound by RTCB residues D119, N354 and K375, and Archease residues K156 and D178 (Supplementary Fig. 4i). Due to the absence of phosphate ions during purification and crystallization, we conclude that the phosphate molecule originates from the pyrophosphate leaving group by subsequent hydrolysis of the β-γ pyrophosphate bond. We observed only minor side-chain movements between the GMPCPP and GMP* structures in the active site.

In the activation complex, Archease blocks access to the active site of RTCB and covers the presumed RNA substrate binding surface entirely. A crystal structure of *Ph*RtcB with a 5′-OH DNA oligonucleotide suggested a potential binding mode for the 3′ exon end of RNA substrates. Superposition with the RTCB-Archease complex revealed a sterical clash between Archease and the bound oligonucleotide (Fig. 4d). Moreover, Y92 undergoes one of the largest side-chain rotations in RTCB upon complex formation, which results in a sterical clash with the suggested 3′ RNA end (Supplementary Fig. 4j)[31]. We conclude that binding to the active site of RTCB by Archease or the RNA substrate is mutually exclusive.

## All metal-coordinating residues are required for guanylylation

The protein–protein interface between RTCB and Archease is mediated by residues in the conserved tip of Archease. Moreover, the Archease residues that interact with the $Mn^{2+}$ and phosphate ions possess the highest degree of conservation. To confirm the importance of these residues, we mutated one metal-coordinating residue, D51, as well as one phosphate-binding residue, K156, to alanine. We first tested their role during RTCB activation by monitoring the incorporation of radioactivity from α³²P-GTP. Both variants abrogated guanylylation of RTCB, indicated by a lacking radioactive signal for the intermediate (Fig. 5a). We further tested the impact on ligation using the *XBP1* intron circularization assay. We could not observe any ligation activity in the presence of either Archease variant (Fig. 5b). Our data confirm the importance of both residues for RTCB guanylylation and—consequentially—for RNA ligation. We conclude that metal coordination by Archease in the activation complex is a prerequisite for the initial guanylylation step.

While the importance of divalent metal ions for RTCB-catalyzed ligation is generally accepted, their precise role and identity is debated[17,23,32,33]. In order to determine the catalytic functionality in the presence of different divalent metal ions, we prepared metal-free RTCB and Archease by using the metal chelator ethylenediaminetetraacetic

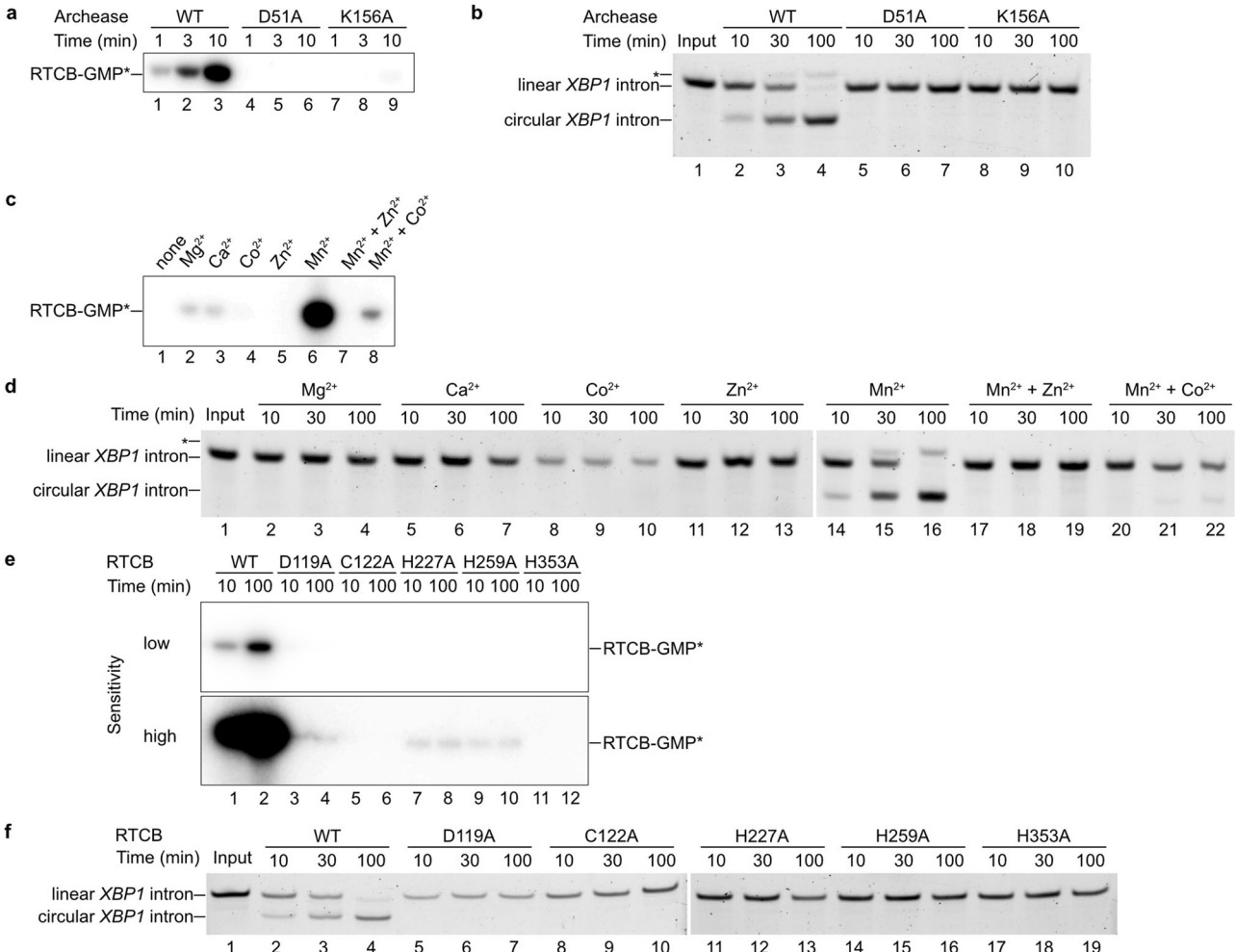

**Fig. 5 | Role of RTCB and Archease active site residues during ligation and guanylylation. a** RTCB (1 μM) guanylylation assay with radioactive α-³²P-GTP (80 nM), Mn²⁺ (0.5 mM), Archease WT and Archease mutants D51A (1 μM) and Archease K156A (1 μM). Aliquots were taken and quenched at different time points, analyzed by urea-PAGE and visualized by autoradiography. All reactions were performed at 30 °C in triplicate (also for (**b–f**)). **b** In vitro ligation of the *XBP1* intron by RTCB (250 nM) with wild-type (WT) Archease (1 μM) or mutants D51A (1 μM) and K156A (1 μM). Aliquots were taken and quenched at different time points and analyzed by denaturing urea-PAGE. The asterisk denotes the guanylylated RNA intermediate in all panels. **c** RTCB guanylylation assay with radioactive α-³²P-GTP and 0.5 mM of the indicated metal ions or mixtures. The assay was performed as

described in (**a**). **d** In vitro ligation of the *XBP1* intron by RTCB (250 nM) and Archease (1 μM) with 0.5 mM of the indicated metal ions. The assay was performed as described in (**b**). Note the use of two separate gels. **e** RTCB guanylylation assay with radioactive α-³²P-GTP (80 nM), Mn²⁺ (0.5 mM), Archease and wild-type RTCB or the indicated RTCB mutant. Phosphor screens were scanned with low and high photomultiplier tube settings, indicated as low and high sensitivity. The assay was performed as described in (**a**). **f** In vitro ligation of the *XBP1* intron by WT RTCB (250 nM) or the indicated RTCB mutant with Archease (1 μM). Ligation reactions were performed as described in (**b**). Note the use of two separate gels. Source data are provided as a Source Data file.

acid (EDTA) (see Methods for details). Without addition of any metal ions to the EDTA-treated proteins, Archease did not promote guanylylation of RTCB (Fig. 5c, lane 1). While Mg²⁺, Ca²⁺ and—to a lesser extent—Co²⁺ showed low levels of covalent incorporation of GMP (Fig. 5c, lanes 2–4), we observed the strongest guanylylation in the presence of Mn²⁺ ions (Fig. 5c, lane 6). We could not detect any guanylylation in the presence of Zn²⁺ (Fig. 5c, lane 5). A previous report showed inhibitory effects on guanylylation by adding equimolar amounts of Zn²⁺ compared to Mn²⁺ on PhRtcB[32,33]. Thus, we tested equimolar mixtures of Mn²⁺ and Zn²⁺ as well as Mn²⁺ and Co²⁺ in RTCB activation assays. In the former case, there was no detectable guanylylation of RTCB (Fig. 5c, lane 7) while the latter displayed diminished guanylylation (Fig. 5c, lane 8). These results are in line with a ligation assay performed upon addition of the tested metal ions (Fig. 5d). We only observed robust ligation in the presence of Mn²⁺, while equimolar amounts of Zn²⁺ and Co²⁺ showed an inhibitory effect.

In the RTCB active site, five conserved residues bind either Mn²⁺(A) or Mn²⁺(B), which are D119, C122, H227, H259 and H353

(Fig. 4c). A potential separation of functions for each metal site was proposed for *Ph*RtcB[19]. Thus, we mutated these metal-coordinating residues to alanine and tested their impact on RTCB guanylylation and RNA ligation. All mutant variants showed impaired activation of RTCB in guanylylation assays. While C122A and H353A displayed no detectable guanylylation, D119A, H227A and H259A exhibited a weak guanylylation signal (Fig. 5e). None of the active site mutants exhibited any ligation activity in *XBP1* intron circularization assays (Fig. 5f). These data suggest that all five metal-binding residues are essential for the ligation reaction due to their role in RTCB-His428 guanylylation.

### The nucleotide-free structure of RTCB reveals conformational plasticity

In all previously reported structures of RTCB-type ligases with a guanine nucleotide present, the guanosine and ribose moieties are deeply buried within the conserved binding pocket. This anchoring of GTP or GMP, respectively, poses the question how the nucleotide is exchanged for a new round of the catalytic cycle. To address the issue of

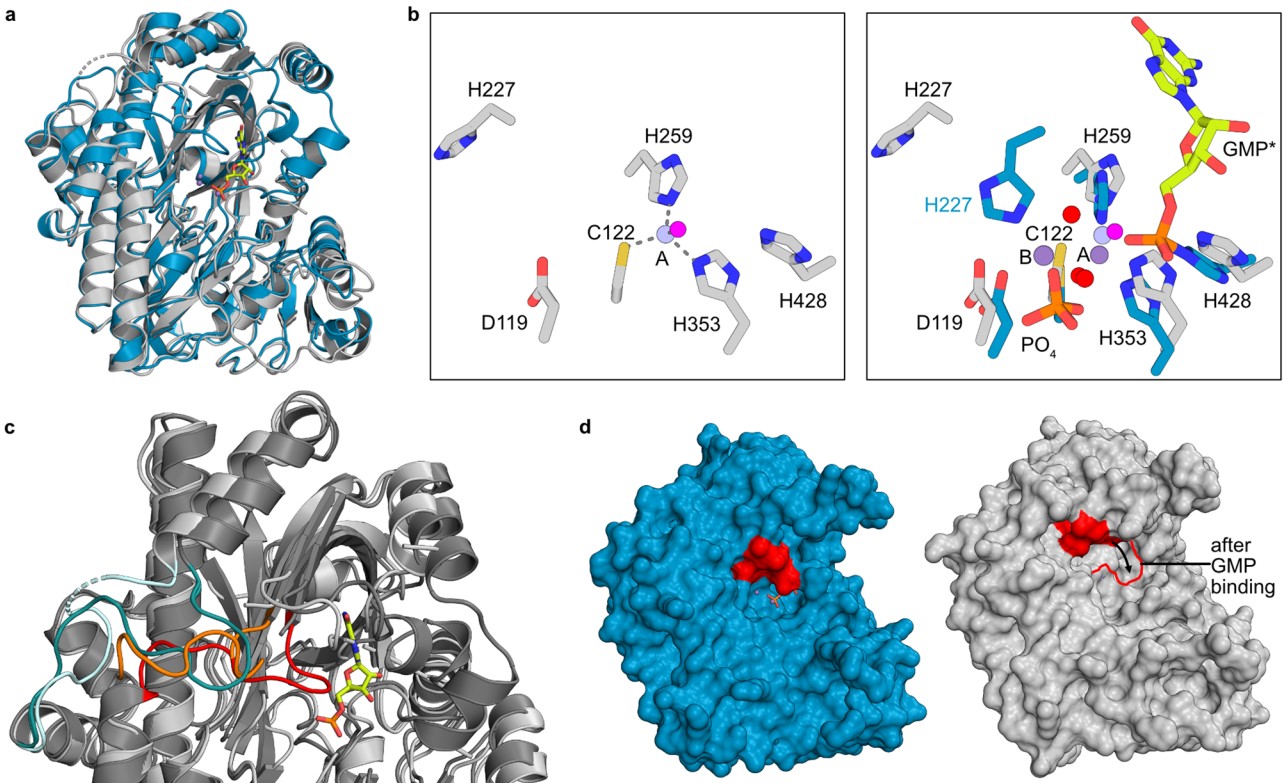

**Fig. 6 | Crystal structure of human RTCB in complex with Mn²⁺.** **a** Superposition of RTCB from the post-activation complex in teal and nucleotide-free RTCB in grey. The Mn²⁺ ions (purple) are depicted as spheres. GMP* (lime) is presented as sticks. **b** Superposition of nucleotide-free RTCB (grey) with the active site of the RTCB from the post-activation complex (teal). The Mn²⁺ ion (purple) and waters (red) from the post-activation complex, the Mn²⁺ ions (light purple) and water (pink) from nucleotide-free RTCB are depicted as spheres. The hydrogen bonds and metal coordination bonds are shown as grey dashed lines. **c** Superposition of RTCB from the post-activation complex (dark grey) and nucleotide-free RTCB (light grey).

GMP* is presented as sticks (lime). The loop region Q218 to Y228 is highlighted in red for the RTCB from the post-activation complex and in orange for nucleotide-free RTCB. The loop region I151 to A165 is highlighted in green for the RTCB from the post-activation complex and in light blue for nucleotide-free RTCB. **d** Surface representation of RTCB from the post-activation complex (teal) on the left and nucleotide-free RTCB (grey) on the right. The loop region Q218 to Y228 is highlighted in red. Loops that were not built for the nucleotide-free RTCB were deleted in the RTCB structure from the post-activation complex for comparison.

nucleotide exchange, we determined the structure of nucleotide-free RTCB with a single Mn²⁺ ion to a resolution of 2.6 Å (Fig. 6a and Supplementary Fig. 5). In contrast to our complex structures, the nucleotide-free RTCB structure contains only one Mn²⁺ ion in position A, coordinated by residues C122, H259, H353 and one water molecule in tetrahedral geometry (Fig. 6b). In addition, residues 45–64, which interact with Archease in the activation complex, adopt a different conformation. It should be noted that the engineered His tag and protease cleavage site were still present during crystallization of this RTCB construct. A total of nine residues before Met1 are resolved and contribute to crystal packing. The lack of electron density in position B indicates a vacated metal binding site in the absence of a guanine nucleotide. The conserved residue H227, which coordinates a metal ion in position B, is not involved in metal binding and rotates away from this position compared to all previously solved structures. Concomitantly, the α-phosphate-binding residue N226 is moved out of the nucleotide binding pocket. The displacement of both conserved residues, which is not present in previously solved RTCB structures, is due to a rearranged loop region from Q218 to Y228. The observed rearrangement coincides with a different conformation of another loop (I151 to A165), which is necessary to avoid steric clashes (Fig. 6c). The conformation of both loops in our nucleotide-free structure changes the accessibility to the guanine nucleotide binding pocket (Fig. 6d and Supplementary Movie 1). Hence, the structure of nucleotide-free RTCB provides an additional open conformation of RTCB-type ligases with regard to the active site.

## Discussion

The mechanistic details of RTCB guanylylation by Archease and the structure of the activation complex have remained unknown. In this study, we present structures of Archease, nucleotide-free RTCB as well as the RTCB-Archease complex in the pre- and post-activation state. Our data provide structural and biochemical insights into their interaction and mechanistic interplay. We propose a crucial role of Archease during RTCB activation by the formation of a composite, metal-dependent active site. Furthermore, the RTCB-Archease complex structure suggests a sequential mechanism in which activation by Archease precedes RNA substrate binding and ligation.

Our in vitro reconstitution data suggest that the formation of the RTCB-GMP intermediate for the human tRNA ligase is strictly dependent on Archease. Thus, the RTCB-Archease activation complex takes a central role within the RTCB-catalyzed reaction cycle (Fig. 7). Our crystal structures of the RTCB-Archease activation complex reveal binding of the largely negatively charged Archease into the positively charged active site grove of RTCB in 1:1 stoichiometry. All existing crystal structures of Archease from different organisms show a dimeric assembly through N-terminal strand swapping. These data hint at a potential role of dimerization during RTCB activation[25,26]. Formation of the RTCB-Archease complex, as presented here, is only possible with the monomeric form of Archease due to the sterical clash with a strand-swapped dimer. While our results do not exclude a functional role of dimerization in vivo, they support the monomeric form of Archease as the functionally active state. This notion is confirmed by

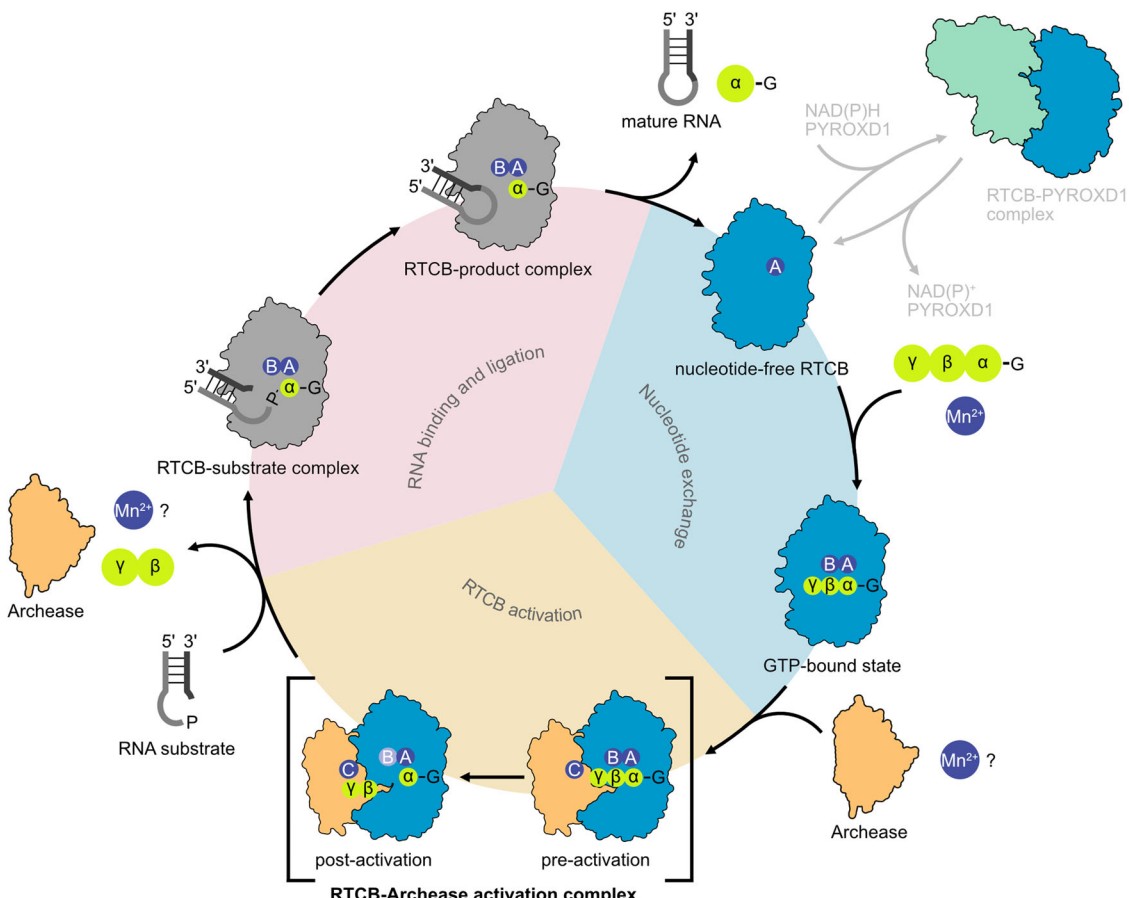

**Fig. 7 | Model of the Archease-dependent, sequential RNA ligation mechanism of RTCB.** The Archease-dependent reaction cycle of the RNA ligase RTCB consists of three phases: nucleotide exchange (1), RTCB activation (2), and lastly RNA binding and ligation (3). To catalyze the RNA ligation reaction, RTCB (teal) relies on two co-factors, GTP (lime) and $Mn^{2+}$ ions (purple, positions are labeled A, B and C). After both co-factors are positioned in the RTCB active site, Archease (orange) binds presumably alongside a third $Mn^{2+}$ ion to RTCB, forming the RTCB-Archease pre-activation complex. Archease enables the formation of the covalent RTCB-His428-GMP intermediate. The presence of $Mn^{2+}$ (B) in the post-activation complex (light purple) is uncertain due to its absence in our structure. The now activated RTCB releases Archease, the pyrophosphate of GTP and possibly $Mn^{2+}$ ions. Binding of either Archease or the RNA substrate to the RTCB active site cavity are mutually exclusive. In the RTCB-substrate complex (grey), the 3′ end of the RNA substrate is activated by the formation of an RNA(3′)-P-P-(5′)G intermediate. The 5′-OH subsequently attacks the activated 3′-P to form a new 3′,5′-phosphodiester bond. The ligated RNA and GMP are released from the RTCB active site allowing the cycle to restart. The RTCB-RNA complexes within the cycle are depicted in grey to indicate missing high-resolution structures. The oxidoreductase PYROXD1 (green) can form an NAD(P)H-dependent complex with RTCB, which protects the active site from oxidation. Archease and PYROXD1 compete for the same interaction surface and bind in a mutually exclusive manner.

our Archease monomer structure, which agrees with a monomeric in-solution NMR structure of an archaeal Archease[28].

Binding of Archease introduces only subtle conformational changes to RTCB, but rather contributes to the active site by coordination of the metal and nucleotide co-factors. In the complex, Archease provides an additional metal binding site (position C) and coordinates the γ-phosphate of the GTP co-factor. Comparison with the structure of *Ph*RtcB with the bound nucleotide analog GTPαS (PDB ID 4ISZ) reveals a similar geometry and binding pattern for the nucleoside and α-phosphate part[18]. In the light of our biochemical results, we speculate that the additional metal-binding site in position C and the resulting positioning of GTP phosphates explains the crucial role of Archease during RTCB guanylylation. The Archease residues involved in metal binding are highly conserved. Moreover, they bind metal ions in two previously reported crystal structures of bacterial and archaeal origin[25,26]. While the $Mn^{2+}$ ions in positions A and C are present in both, the pre- and post-activation complex, position B is only occupied by $Mn^{2+}$ in the pre-activation state. Interestingly, position B is unoccupied in our nucleotide-free RTCB monomer structure as well as in several structures of *Ph*RtcB. The emerging pattern is that the metal ion in position B is less tightly bound when a guanosine nucleotide is lacking or when the active site histidine is guanylylated. Thus, it is missing in several such structures[18,19]. It remains to be elucidated if the activation process requires the presence of two or three metal ions. The former scenario would imply that the metal ion in position B switches to position C during activation, which leaves the possibility of ion removal upon dissociation of Archease. In the latter case, Archease already contains a pre-bound metal ion, which it provides as the third active site metal center by binding to RTCB. We cannot distinguish between both scenarios based on our current results. In our complex structures, the $Mn^{2+}$ ion in position C is in the vicinity of the γ-phosphate. We speculate that its function is to facilitate the nucleophilic attack by H428 on the α-phosphate by correct positioning of all phosphate groups and stabilization of the pyrophosphate leaving group. In the post-activation complex, only a single phosphate is present instead of the expected pyrophosphate, which is the leaving group during guanylylation of RTCB. We hypothesize that hydrolysis of the pyrophosphate in the crystal resulted in the observed monophosphate. It is not clear if further hydrolysis of the β-γ anhydride bond was only due to the prolonged crystallization process or

indeed part of the activation mechanism. Thus, the Archease-coordinated metal ion in position C could either promote removal of the pyrophosphate from the active site or even enable further hydrolysis to drive the reaction.

While the presence of divalent metal cations is essential for RTCB activation, and thus for RNA ligation, the identity of the bound metal cations, particularly in vivo, remains uncertain. In our in vitro reconstitution, we identified $Mn^{2+}$ as the canonical and preferred metal cation and detected a competing effect for $Co^{2+}$ and $Zn^{2+}$. The lack of guanylylation in the presence of $Zn^{2+}$ and the competing effect with $Mn^{2+}$ were previously observed for *E. coli* and *P. horikoshii* RtcB[17,32,33]. The general argument is that zinc as a softer Lewis acid binds stronger than manganese due to the conserved cysteine in the active site while it does not support guanylylation chemistry. With regards to $Co^{2+}$, the various studies came to different conclusions. $Co^{2+}$ ions promote guanylylation in *P. horikoshii* and human RTCB, while *E. coli* RTCB cannot use $Co^{2+}$ for activation[17,23,32,33]. The described trace levels of guanylylation and complete lack of ligation with $Co^{2+}$ in our study is at odds with the previous results for human RTCB, which reported the strongest activity in the presence of $Co^{2+}$ ions[23]. We recognize that the possibility of more than one type of metal ion in the active site under physiological conditions and in the context of the holo-complex will require further exploration.

Our structural data provide insights into the order of events during the tRNA-LC reaction cycle at the following steps: (1) the guanine nucleotide exchange in between two cycles, (2) formation of the activation complex and guanylylation, as well as (3) Archease release followed by RNA binding and ligation (Fig. 7). Our structure of the RTCB monomer shows evidence for a conformational change of two loop regions, I151 to A165 and Q218 to Y228, in the nucleotide-free state. The concerted movement of both loops suggests a latching motion towards the GTP molecule during the initial nucleotide binding step prior to guanylylation. Strikingly, the conformation of the aforementioned loops in the open state would sterically clash with binding of Archease. On the one hand, the open, nucleotide-free state precludes Archease binding before RTCB activation. On the other hand, the binding of Archease presumably prevents nucleotide exchange in the activation complex. Thus, we propose that GTP binding facilitates the interaction with Archease. In addition to precluding the nucleotide binding pocket, Archease blocks the access for RNA substrates to the RTCB active site. Further experiments are needed to address preferential interaction modes with Archease and RNA depending on RTCB's nucleotide state. Following activation and Archease release, the guanylylated active site is available for productive binding of the RNA substrate allowing completion of the final exon-exon ligation step. In conclusion, we surmise that the conformational changes associated with nucleotide binding as well as the mutually exclusive binding of Archease and RNA ensure the correct sequence during the catalytic cycle of the tRNA-LC.

While the RTCB-Archease structures provide a detailed understanding of the ligase activation complex, we have an incomplete understanding of the dynamics of complex formation and post-guanylylation dissociation. We cannot fully recapitulate the seemingly transient nature of the RTCB-Archease interaction in solution considering the charge complementarity and extensive protein-protein surface as revealed by our structures. While one previous study determined an in vitro dissociation constant in the low nM range[24], others were not able to detect a direct interaction between both proteins[23,27]. There are two reports of successful co-immunoprecipitation of RTCB and Archease[26], albeit one required crosslinking for stabilization[24]. These previous studies did not reveal an effect of co-factors on the RTCB-Archease interaction. Our in vitro crosslinking results hint at preferential binding to the GTP-bound and occupied metal site(s) state of RTCB. Additional studies dissecting the preferred conditions and molecular discrimination mechanisms for

RTCB-Archease complex formation will be needed. At the stage of the post-activation complex, one major question remains unanswered: what triggers the release of Archease following guanylylation of the active site histidine? Our complex structure in the guanylylated post-activation state does not show major conformational rearrangements within RTCB. It is conceivable that the shift of one metal ion from position B to C and the concomitant release of the pyrophosphate weaken the RTCB-Archease interaction enough to trigger the release. The arriving RNA substrate might promote the release via direct displacement, but currently there is no evidence for or against such a role.

We surmise that our key findings about Archease-mediated activation of RTCB hold true in the context of the tRNA ligase holo-complex. However, the other core components, DDX1, FAM98B, CGI-99 and ASW, might affect binding of Archease and thus play an unforeseen role during the activation process. Intriguingly, DDX1 was shown to enhance Archease-dependent guanylylation[24]. The involvement of the additional subunits in the context of the RTCB reaction cycle requires further studies. Javier Martinez and co-workers identified two protein factors that play a crucial role for RTCB function: Archease, which we dissected in this study, and the redox regulator PYROXD1, which protects the RTCB active site from oxidative inactivation[34]. A complementary study presents the structure of the RTCB-PYROXD1 complex[35]. The—apparently—mutually exclusive binding of both activity regulators necessitates a permanent sampling of RTCB's active site in the cellular context (Fig. 7). Together, these factors have to interplay to allow both, efficient enzymatic turnover and protection from oxidation. Retaining the essential role of RTCB in tRNA splicing from archaea to metazoa resulted in an increased number of molecular interactions (or additional players, like PYROXD1). It remains to be seen if additional regulatory mechanisms or specific cellular functions of metazoan RTCB will be uncovered that rely on the multiplicity of molecular interactions.

In conclusion, this work advances our understanding of Archease-mediated activation of the essential human tRNA ligase complex and other RTCB-type ligases. We provide biochemical and structural insights into the direct role of Archease during RTCB-catalyzed RNA ligation by coordinating the GTP and metal co-factors. Our study provides structural snapshots of RTCB and Archease along the tRNA-LC reaction cycle. We postulate a sequential reaction cycle, in which nucleotide exchange at the start of the cycle, followed by Archease-dependent activation and the final RNA substrate processing are the result of RTCB conformational changes and complex formation with the respective interaction partners.

## Methods
### Site-directed mutagenesis
Archease mutants D51A, K156A and S147C and RTCB mutants D119A, C122A, H227A, H259A and H353A were cloned using site-directed mutagenesis (Q5® Site-Directed Mutagenesis Kit, NEB) using primers TCATACAGCAGCGGTCCAGTTAC (forward) and TCCAAATACTCGTA CTTC (reverse) for Archease D51A and AACAGAAGTCGCAGCAAT AACATATTCAGCAATGC (forward) and CCCTGAGGGTGCTTGGAC (reverse) for Archease K156A, ATTTTCATTGTGTAAGCACCCTCAGG (forward) and TCTTCTCCCCACCCAATTG (reverse) for Archease-S147C, TGTCGGGTTTGCCATCAACTGTG (forward) and CCACCT GGGGATACTACT (reverse) for RTCB D119A, (forward) and (reverse) for C122A, AGCAGGCAACGCTTATGCAGAAATCCAGGTTG (forward) and CCCAGGGTCCCCAACTGA (reverse) for RTCB H227A and TGT GATGATCGCCAGTGGAAGCAGAG (forward) and CACACCTGTCCCTT ATGG (reverse) for RTCB H259A and TGATGTTTCTGCCAACATTG CCAAAG (forward) and TAGATCACATGTAGGTCC (reverse) for RTCB H353A. Mutant variants were expressed and purified as the respective wild-type proteins (see below).

## Expression and purification of recombinant human Archease

Human Archease isoform 1 (UniProt A8K0B5; note the deviating translation start sites from UniProt Q8IWT0 by twelve additional residues) was expressed and purified as described previously[10]. In short, the fusion protein with an N-terminal His$_6$ tag followed by a thrombin protease cleavage site was expressed in *E. coli* BL21-CodonPlus (DE3)-RIPL (Agilent Technologies) cells. The clarified lysate was applied to a 5 mL Ni-NTA column (Cytiva). After tag removal (optional) with thrombin (Cytiva), the sample was re-applied onto the Ni-NTA column to remove the affinity tag. The flow-through fraction was concentrated using a centrifugal filter (Amicon Ultra, MWCO 10 kDa, Sigma) and purified by size-exclusion chromatography with buffer containing 25 mM HEPES, pH 7.5, 100 mM NaCl$_2$, 5% Glycerol, 1 mM TCEP on HiLoad16-600 Superdex 75 pg column (Cytiva). Archease containing peak fractions were concentrated, flash-frozen in liquid nitrogen and stored at −80 °C. We determined a 260/280 nm ratio of 0.56 for the final protein preparation indicating the absence of nucleic acid contaminants (see Source Data).

## Insect cell culture and expression of human RTCB

*Spodoptera frugiperda* (*Sf*21) cells (kind gift from the Jeske Lab) were cultivated at 27 °C and 120 rpm in Sf-900™ III SFM medium (Thermo Fisher Scientific). Cells were maintained in exponential growth and diluted by passaging to 0.3–0.5 × 10$^6$ cells/mL every 2 or 3 days. Recombinant baculoviral BACs were generated by Tn7 transposition in *Escherichia coli* DH10EMBacY cells (Geneva Biotech) and the presence of the *RTCB* gene was verified by PCR. To prepare a first generation of virus (V$_0$), *Sf*21 cells from a suspension pre-culture were seeded into six-well-plates at a cell density of 0.8 × 10$^6$ cells/mL per well. 10 μg of the isolated recombinant bacmid and 5 μL Fugene HD transfection reagent (Promega) were diluted in 150 μL Sf-900™ III SFM. After 15 min of incubation at room temperature, the mix was added to the culture media covering the adherent cells. The virus supernatant V$_0$ was harvested 66–72 h post transfection. Recombinant initial baculovirus (V$_0$) was harvested from the cell supernatant and used for the production of amplified baculovirus (V$_1$) in *Sf*21 suspension cultures. RTCB protein was produced in 500 mL of *Sf*21 suspension culture at a cell density of 0.9–1.2 × 10$^6$ cells/mL by infection with 1% (v/v) of V$_1$ baculovirus supernatant. 66–80 h post cell proliferation arrest, insect cells were harvested by centrifugation at 500×*g* for 10 min. After PBS wash, cell pellets were flash-frozen in liquid nitrogen and stored at −80 °C until further use. Every course of baculoviral infection was monitored by increase in cell diameter, cell growth as well as YFP fluorescence, since the EMBacY backbone contains a constitutively expressing YFP expression cassette which allows monitoring of viral titers via fluorescence. Protein expression was analyzed by 12% SDS-PAGE.

## Purification of recombinant human RTCB

Harvested *Sf*21 cells were lysed via microfluidizer in lysis buffer containing 25 mM Tris, pH 7.8, 300 mM NaCl, 20 mM imidazole, 10% glycerol and 1 mM TCEP. The lysate was clarified by centrifugation for 20 min at 20,000× *g* at 4 °C. The clarified lysate was applied to a 5 mL Ni-NTA column (Cytiva), the cartridges were washed with the same lysis buffer, and bound protein was eluted with buffer containing 25 mM Tris, pH 7.7, 300 mM NaCl, 500 mM imidazole, 10% glycerol and 1 mM TCEP. The fusion tag was removed (optional) by His$_6$-TEV protease during overnight dialysis at 4 °C in buffer containing 25 mM Tris, pH 7.8, 200 mM NaCl, 10% glycerol, 1 mM DTT and 250 μM EDTA. The sample was subsequently re-applied onto the Ni-NTA to remove the affinity tag and His$_6$-TEV protease. The flow-through fraction was diluted 1:5 with AEX dilution buffer (25 mM Tris-HCl, pH 7.85, 10% glycerol and 1 mM TCEP). The sample was subsequently applied onto the anion exchange column, HiTrap HP Q (Cytiva). Fractions containing RTCB were pooled, concentrated using a centrifugal filter (Amicon Ultra, MWCO 30 kDa, Sigma) and purified by size-exclusion chromatography (HiLoad 16/600 Superdex 200 pg, Cytiva), eluting with buffer containing 25 mM Tris, pH 7.5, 200 mM NaCl, 5% glycerol and 1 mM TCEP. RTCB-containing peak fractions were concentrated using a centrifugal filter (Amicon Ultra, MWCO 30 kDa, Sigma), flash-frozen in liquid nitrogen and stored at −80 °C. We determined a 260/280 nm ratio of 0.55 for the final protein preparation indicating the absence of nucleic acid contaminants (see Source Data).

## Crystallization

For crystallization of Archease, the protein was concentrated using a centrifugal filter (Amicon Ultra, MWCO 30 kDa, Sigma) to 41.5 mg/mL. Crystals were grown using sitting drop vapor diffusion at 291 K. Crystals appeared after 10 days in 0.2 M ammonium chloride, 20% (w/v) PEG 3350 and were flash-frozen in liquid nitrogen using ethylene glycol as cryoprotectant.

For crystallization of the RTCB-Archease post-activation complex, the proteins were concentrated using centrifugal filters (Amicon Ultra, MWCO 30 kDa and MWCO 10 kDa, Sigma) to 10 mg/mL (RTCB) and 47.1 mg/mL (Archease with N-terminal His$_6$ tag) and mixed in a 1.25:1 ratio (Archease:RTCB). The sample contained 2 mM GTP and 2 mM MnCl$_2$. Crystals were grown using sitting drop vapor diffusion at 291 K. A crystal appeared after 5 days in 10% (w/v) PEG 8000, 0.1 M HEPES pH 7.5, 8% (v/v) ethylene glycol and were flash-frozen in liquid nitrogen using ethylene glycol as cryoprotectant.

For crystallization of the RTCB-Archease pre-activation complex, the proteins were concentrated to 10 mg/mL (RTCB) and 41.5 mg/mL (Archease) mixed in a 1.25:1 ratio (Archease:RTCB). The sample contained 1 mM GMPCPP and 2 mM MnCl$_2$. Crystals were grown using sitting drop vapor diffusion at 291 K. Crystals appeared after 20 days in 0.2 M potassium nitrate, 20% w/v PEG 3350 and were flash-frozen in liquid nitrogen using ethylene glycol as cryoprotectant.

For crystallization of nucleotide-free RTCB, the protein (with N-terminal His$_6$ tag) was concentrated using a centrifugal filter (Amicon Ultra, MWCO 30 kDa, Sigma) to 8.1 mg/mL. Crystals were grown using hanging drop vapor diffusion at 291 K in 0.1 M magnesium chloride, 0.1 M HEPES, pH 7.0, 15% PEG 4000 in the presence of 2 mM manganese chloride and were flash-frozen in liquid nitrogen using ethylene glycol as cryoprotectant.

## Data collection and structure determination and analyses

Data sets were collected at cryogenic temperature for Archease and RTCB-Archease complex at ESRF beamline ID23-1 and for RTCB at APS Argonne beamline 23ID-D. X-ray fluorescence (XRF) spectra were recorded to determine the identity of the observed metal sites as implemented at ESRF beamlines[36] (see Source Data). Diffraction images were integrated with XDS[37] and then scaled using AIMLESS[38] as part of the CCP4i software package[39]. The resolution cut-offs were selected based on the half-data set correlation CC$_{1/2}$ as implemented in AIMLESS[40]. Phases were obtained by molecular replacement with PHASER-MR[41] implemented in the PHENIX package[42]. Residue 53–180 of human Archease (PDB ID 5YZ1) served as search model for Archease. The model of human RTCB (UniProt ID Q9Y3I0) from the AlphaFold database[43] was used as search model for RTCB. Iterative model building and refinement was performed with Coot[44] and Phenix.refine[45]. The quality of the resulting structural models was analyzed with MolProbity[46]. Structure figures were prepared with PyMOL 2.4.1 (The PyMOL Molecular Graphics System, Schrödinger, LLC.). Crystallographic data are summarized in Table 1. Coordinates and structure factors are deposited at the Protein Data Bank PDB with accession codes 8BTX (Archease), 8BTT (RTCB), 8ODP (RTCB-Archease pre-activation complex) and 8ODO (RTCB-Archease post-activation complex). Multiple sequence alignments were performed using T-Coffee's Expresso[47]

**Table 1 | Crystallographic data collection and refinement**

| | Archease | RTCB-Archease complex, pre-activation state | RTCB-Archease complex, post-activation state | RTCB |
|---|---|---|---|---|
| PDB ID | 8BTX | 8ODP | 8ODO | 8BTT |
| **Data collection** | | | | |
| Space group | P 4₂ 2 2 | P 1 2₁ 1 | P 1 2₁ 1 | I 4 |
| Resolution (Å) | 79.98–1.84 (1.88–1.84) | 122.60–2.30 (2.34–2.30) | 119.23–2.20 (2.24–2.20) | 48.22–2.60 (2.72–2.60) |
| Unique reflection | 18,949 (1137) | 124,773 (6124) | 145,859 (6868) | 33,777 (4030) |
| Cell dimensions | | | | |
| a, b, c (Å) | 79.98, 79.98, 65.16 | 98.43, 122.60, 124.76 | 99.09, 125.55, 125.73 | 215.64, 215.64, 46.87 |
| α, β, γ (°) | 90, 90, 90 | 90, 107.88, 90 | 90, 108.51, 90 | 90, 90, 90 |
| $R_{merge}$ | 0.068 (3.698) | 0.095 (1.161) | 0.110 (1.160) | 0.184 (2.150) |
| $R_{pim}$ | 0.013 (0.743) | 0.039 (0.465) | 0.045 (0.495) | 0.077 (0.875) |
| Mean (I/σ(I)) | 23.7 (1.0) | 11.4 (1.5) | 9.2 (1.1) | 8.5 (0.9) |
| Multiplicity | 27.5 (25.1) | 7.0 (7.3) | 7.0 (6.3) | 6.7 (7.0) |
| Completeness (%) | 100 (100) | 99.8 (99.5) | 99.3 (95.2) | 100 (100) |
| $CC_{1/2}$ | 0.999 (0.471) | 0.998 (0.786) | 0.998 (0.860) | 0.995 (0.348) |
| **Refinement** | | | | |
| $R_{work}$ (%) | 19.5 | 18.1 | 19.3 | 19.9 |
| $R_{free}$ (%) | 21.6 | 21.6 | 21.8 | 25.7 |
| RMSD Bond length (Å) | 0.007 | 0.007 | 0.005 | 0.003 |
| RMSD Bond angle (°) | 0.850 | 0.800 | 0.630 | 0.530 |
| Ramachandran favored (%) | 97.44 | 96.78 | 96.97 | 96.23 |
| Ramachandran allowed (%) | 2.56 | 2.92 | 2.85 | 3.56 |
| Ramachandran outliers (%) | 0 | 0.30 | 0.19 | 0.21 |
| Rotamer outliers (%) | 0 | 0.31 | 0.67 | 0.63 |
| Clash score | 2.72 | 6.68 | 2.92 | 4.08 |
| Average B factor (Å²) | | | | |
| Protein | 58.3 | 63.3 | 65.5 | 71.4 |
| Ligands | – | 57.1 | 57.2 | 90.2 |
| Solvent | 60.4 | 53.5 | 56.8 | 60.4 |

Values in parentheses refer to the highest resolution shell.
The $R_{free}$ set consists of 5% randomly chosen data excluded from refinement.

and were visualized with ESPript 3.0 (https://esprit.ibcp.fr)[48]. Conservation analysis was performed using ConSurf [29,30]. PISA[49] and PLIP[50] were used for interaction analyses.

### Labeling with Alexa Fluor maleimide dyes
AF488 maleimide or AF647 maleimide (Jena Bioscience) was added in a tenfold molar excess to Archease-S147C (50 μM) or RTCB (50 μM), respectively. RTCB was labeled in the presence of $Mn^{2+}$ (500 μM) and GMPCPP (500 μM). After 2 h incubation at room temperature, the reaction was stopped with an excess of β-mercaptoethanol. The sample was applied onto a desalting column (HiTrap Desalting, Cytiva) and eluted with desalting buffer (25 mM HEPES, pH 7.3, 250 mM NaCl₂, 5% Glycerol, 1 mM TCEP) to remove excess dye. Labeled Archease-S147C or RTCB was flash-frozen in liquid nitrogen and stored at 80 °C, protected from light.

### Radioactive guanylylation assay
Depending on the experimental setup, 80 nM, 399.6 kBq α³²P-GTP (Hartmann Analytic GmbH), 0.5 mM MnCl₂, 0.5 mM CoCl₂, 0.5 mM ZnCl₂, 0.5 mM CaCl₂, 0.5 mM MgCl₂, 1 μM RTCB or 1 μM Archease in reaction buffer (20 mM HEPES, pH 7.5, 70 mM NaCl, 1 mM TCEP, 5% glycerol) was incubated at 30 °C. The reaction was stopped at different time points with 4× loading dye (1 M tris, pH 6.8, 350 mM SDS, 50% glycerol, 3.7 mM bromophenol blue, 25% β-mercaptoethanol). The samples were separated on a 12% SDS-PAGE gel (nuPAGE, Invitrogen by Thermo Fisher Scientific) and the exposed phosphor screens were scanned using an FLA-7000 (FUJIFILM).

### In vitro splicing/ligation assay
A cleavage reaction, using IRE1-KR (containing IRE1α's kinase and RNase domains) prior to the ligation assay, was performed for the substrate *XBP1u* but not for the *XBP1* intron. The *XBP1u* RNA was diluted with buffer containing 20 mM HEPES, pH 7.5, 100 mM NaCl, 1 mM MgCl₂. The RNA was unfolded at 90 °C for 1 min and refolded at room temperature (let cool down to ~40 °C). 0.5 μM IRE1-KR and 0.5 μM *XBP1u* were mixed in IRE1-KR cleavage reaction buffer (20 mM HEPES, pH 7.5, 70 mM NaCl, 2 mM MgCl₂, 1 mM TCEP, 5% glycerol). The cleavage reaction was performed at 37 °C for 105 min. The ligation reaction mix consisted of 500 nM/250 nM (for *XBP1* intron ligation) RTCB, 1 μM Archease, 250 nM *XBP1u* after cleavage or 1 μM *XBP1* intron, 1 mM GTP or ATP, 0.5 mM MnCl₂ in cleavage reaction buffer (20 mM HEPES, pH 7.5, 70 mM NaCl, 2 mM MgCl₂, 1 mM TCEP, 5% glycerol) and was incubated at 30 °C. Samples were taken at different time points and the reaction was quenched with 10× stop solution (10 M urea, 0.1% SDS, 1 mM EDTA, trace amounts of xylene cyanol and bromophenol blue). RNA samples were unfolded at 80 °C for 2 min and subsequently placed on ice to prevent refolding. The samples were analyzed by 15% denaturing urea-PAGE and visualized via SYBR™ Gold Nucleic Acid Gel Stain (Invitrogen by Thermo Fisher Scientific).

### Metal specificity of guanylylation and ligase activity
RTCB or Archease were incubated for 1 h on ice with 10 mM EDTA in buffer (20 mM HEPES, pH 7.5, 1 mM TCEP, 5% glycerol). EDTA was removed using a protein desalting spin column (Thermo Fisher

Scientific) pre-equilibrated with buffer (20 mM HEPES, pH 7.5, 70 mM NaCl, 1 mM TCEP, 5% glycerol). The metal ligation assays were performed as described above for ligation assays with the exception of the reaction buffer composition 20 mM HEPES, pH 7.5, 70 mM NaCl, 1 mM TCEP, 5% glycerol) and the indicated metal ions (0.5 mM).

### BS3 crosslinking assay
In total, 50 μM BS3 was mixed with varying components, depending on the experiment: 10 μM RTCB, 10 μM Archease, 1 mM GTP and 500 μM MnCl$_2$ in conjugation buffer (20 mM HEPES, pH 7.5, 70 mM NaCl, 2 mM MgCl$_2$, 5% glycerol, 1 mM TCEP). For the fluorescence crosslinking assay, 5 μM RTCB-AF488 and 5 μM Archease-S147C-AF647 were used. The reaction mix was incubated for 15 min at room temperature and quenched with quenching buffer (1 M Tris, pH 7.5, 1 M glycine) for 15 min at room temperature. The proteins were separated on a 12% SDS-PAGE gel and stained with Coomassie blue.

### Polyacrylamide gel electrophoresis
Splicing and ligation reactions were carried out as described above, and the reaction products were analyzed by 15% urea-PAGE using stop solution (10 M urea, 0.1% SDS, 1 mM EDTA, 0.1% xylene cyanol, 0.1% bromophenol blue). The gels were run in 1x TBE-buffer at 150 V (constant voltage) at room temperature for 70 min. The gels were subsequently stained with SYBR Gold nucleic acid stain (Invitrogen, Life Technologies) to visualize the RNA by ultraviolet trans-illumination.

### Sodium dodecyl sulfate polyacrylamide gel electrophoresis
Guanylylation and crosslinking reactions were carried out as described above, and the reaction products were analyzed by 12% SDS-PAGE using 4x loading dye (1 M tris, pH 6.8, 350 mM SDS, 50% glycerol, 3.7 mM bromophenol blue, 25% β-mercaptoethanol). The gels were run in 1x TGS buffer (248 mM Tris, 1.92 M glycine and 1% SDS) at 200 V (constant voltage) at room temperature for 50 min. The gels were subsequently stained with Coomassie blue to visualize the proteins.

### Statistics and reproducibility
All gel-based experiments were performed in triplicate. No statistical method was used to predetermine sample size. No data were excluded from the analyses. The experiments were not randomized and the investigators were not blinded to allocation during experiments and outcome assessment.

### Reporting summary
Further information on research design is available in the Nature Portfolio Reporting Summary linked to this article.

## Data availability
Structure factors and atomic coordinates of the reported X-ray crystallographic structures have been deposited in the Protein Data Bank (PDB) with accession codes: 8BTT (RTCB, nucleotide-free), 8BTX (Archease), 8ODO (RTCB-Archease complex with GMP), 8ODP (RTCB-Archease complex with GMPCPP). For structural analysis, we have used the AlphaFold model of human RTCB from UniProt (ID Q9Y3I0) [https://alphafold.ebi.ac.uk/entry/Q9Y3I0]. We have also compared and superimposed our structural models with the following PDB entries: 5YZ1, 7P3B, 4ISZ and 7LFQ. Source data are provided with this paper.

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

## Acknowledgements

We thank Sandra Köhler for experimental assistance, Rafael Salazar for help with the ConSurf analysis, and Jutta Metz for advice on insect cell expression. We are thankful to Martin Jinek, Javier Martinez, Luuk Loeff and Igor Asanovic for helpful discussions and sharing of unpublished data. We acknowledge the European Synchrotron Radiation Facility (ESRF) for provision of synchrotron radiation facil-ities and we would like to thank Nicolas Coquelle and Alexander Popov for assistance and support in using beamline ID23-1. We thank the entire staff at GM/CA@APS, which has been funded in whole or in part with Federal funds from the National Cancer Institute (ACB-12002) and the National Institute of General Medical Sciences (AGM-12006). This research used resources of the Advanced Photon Source, a US Department of Energy (DOE) Office of Science User Facility operated for the DOE Office of Science by Argonne National Labora-tory under Contract No. DE-AC02-06CH11357. We thank the 2016 CCP4 School at Argonne including all its instructors for help and guidance to JP during data collection. We acknowledge support from the BZH Crystallization Platform and thank Claudia Siegmann for help with protein crystallization. The authors gratefully acknowledge the data storage service SDS@hd supported by the Ministry of Science, Research and the Arts Baden-Württemberg (MWK) and the German Research Foundation (DFG) through grant INST 35/1314-1 FUGG and INST 35/1503-1 FUGG. S.M. acknowledges a scholarship from the Mexican Council for Science and Technology (CONACYT) and the German Academic Exchange Service (DAAD). J.P. acknowledges funding from the Deutsche Forschungsgemeinschaft (DFG, German Research Foundation)—Emmy Noether Program (project number 442512666) and TRR319 RMaP TP-A06 (project number 439669440).

## Author contributions

J.L.G. performed most experiments and analyzed data. S.I.M.G. and L.W. performed experiments. P.H. assisted with insect cell expression and provided materials. J.K. analyzed crystallographic data. J.P. conceived the study, performed experiments and analyzed data. J.L.G., J.K. and J.P. wrote the manuscript with contributions from all authors.

## Funding

## Competing interests

The authors declare no competing interests.
