## [Peer Review File · Nature Communications]

Structural and mechanistic insights into activation of the human RNA ligase RTCB by ArcheaseReviewer #1 (Remarks to the Author):

In this work, Gerber and colleagues provide the mechanistic basis for the activation of the human RTCB-Archease complex, by imaging the complex in its pre- and post-activation states. Their work is complemented by biochemical work, where the authors also reconstitute the ligation reaction in vitro.

RTCB-type ligases are present in all domains of life and, like canonical Trl1-type ligases, they ligate RNA ends carrying a 2',3'-cyclic phosphate and a 5'-OH. In tRNAs these ends are generated by TSEN complex. In addition, the tRNA ligase complex also ligates XBP1 mRNA, which is a crucial component of the unfolded protein response (UPR).

Human RTCB is an integrated part of the pentameric human tRNA ligase complex, which also harbors of DDX1, CGI-99, FAM98B and ASHWIN. The ligation reaction by the tRNA ligase complex is known to require GTP-dependent activation of RTCB. During the reaction RTCB undergoes guanylylation, which depends on the on an activation factor, named Archease.

First, the authors reconstitute the ligation reaction of XBP1 mRNA fragments after cleavage by IRE1, using purified human RTCB and Archease proteins. They further show that the multi-turnover reaction as well as the interaction between the two protein partner depend on the presence of GTP and Mn. Next, they determine several high-resolution crystal structures of human Archease, RTCB and the RTCB-Archease complex in its pre- and post-activation states. They use these structures to compare the conformation of the active site residues and validate their structural findings with structure-guided mutants.

The work closes several important knowledge gaps by structurally resolving important intermediates of the reaction. In conclusion, I would like to congratulate the authors for their work and I support publication of the manuscript in Nature Communications after resolving a few minor issues.

Obviously, there is one major remaining issue that is not touched experimentally at all - how is the mechanism incorporated in the fully assembled human tRNA ligase complex and does it work slightly different for tRNA substrates. However, the presented work provides a solid mechanistic basis to precisely define those next challenges, which definitely need to be addressed in the future.

Minor issues

- Please define the RTCB abbreviation for the non-expert reader
- I feel that the last sentence of the abstract is an overstatement "Taken together, we present structural snapshots along the reaction cycle of the tRNA ligase complex." Please rephrase accordingly – the work does not provide any experimental evidence that the mechanisms works identically within the ligase complex – even if one can assume it is the case.
- "in vitro" should be italic
- Page 8 – line 143 – "Based on the results of the crosslinking experiments,...". I am slightly confused, if the crystallization was performed using cross-linked complexes – from the M&M section, it seems that the proteins were just used at high concentrations and no cross-linker was used. Please rephrase the text to avoid any confusion. Please mention if any reconstitution experiments were performed and if the complex did not withstand any gel filtration approaches (which is not uncommon for dynamic complexes).
- In Extended Data Fig. 1, the authors show a SDS-PAGE gel of the purified proteins – it would be good to also show the Absorbance profile to show that both proteins did not purify with any bound RNA contaminants. Could such bound RNA molecules be the reason for problems to reconstitute the complex in vitro? Was the sample ever applied to a Heparin column to remove bound Nucleic acids?

- The authors conclude that all observed sites are Mn sites – were any efforts undertaken to measure and calculate anomalous difference Fourier maps at Mn-specific wavelengths? If not, please make a comment about the remaining uncertainty of the identity of the atom. Mg could have been co-purified during the purifications. Same applies to the phosphate atoms of the potential pyrophosphate in the guanylylated post-activation complex.

Sebastian Glatt

Reviewer #2 (Remarks to the Author):

The article 'Structural and mechanistic insights into activation of the human RNA ligase RTCB by Archease' by Gerber et al. reports on various structural snapshots of human RTCB, the active component of the human tRNA ligase complex, during its catalytic cycle. Besides crystal structures of human Archease and RTCB in the apo state, the authors captured a transient ternary complex of RTCB, Archease, and the GTP analog GMPCPP and solved its crystal structure. Furthermore, they present a crystal structure of activated, guanylylated RTCB in complex with Archease. The authors corroborate their structural data with functional analyses that shed light on the metal ion dependency of RTCB guanylylation catalyzed by Archease and the Archease dependence of RTCB-catalyzed RNA ligation. The clear novelty of the work by Gerber et al. lies in, first, capturing transient chaperoning and activating states of the human transfer RNA ligase RTCB, and second, in determining structural snapshots of these states that significantly augment our understanding of RTCB-catalyzed RNA ligation.

In my opinion, the manuscript will benefit from a restructured presentation of the data focusing on RTCB rather than Archease. Therefore, the authors should consider introducing and describing their crystal structure of the ternary RTCB-Archease-GMPCPP complex in the pre-activated state first, then presenting the activated state of RTCB, and eventually discuss the structures of the individual components including the structural dynamics that are seen upon complex formation—in a similar order as written in their abstract.

Major:

1) The authors should show an overview of the interaction surface(s) between RTCB and Archease in the main text and discuss interacting residues in a separate figure, which may be referred to in line 151 after '...with an area of about 2,200 Å².'. Do the crystal structures explain the transient character of the complex? As the authors provide evidence that fluorescently labelled Archease chemically cross-links to recombinant RTCB, fluorescence anisotropy measurements may be considered for determination of dissociation constants (K_d) of the ternary complex to further define its transient character.

2) Please revisit items that are presented in main Figures and Extended Data Figures (as pointed out below). The paragraph entitled 'Crystal structures of the RTCB-Archease complex in two distinct nucleotide-bound states' should be split in two parts (between lines 185 and 186) to logically separate the descriptions of the pre-activation and the activated state of RTCB. The current paragraph reads as if the authors accidentally obtained the activated state of RTCB. Please substantiate the sentence 'The position of the covalent bond at the α-phosphate was inverted due to the phosphoryl transfer reaction,...' (lines 192-193).

Minor:

1) Consider changing the title of Figure 1 as the data refers to a functional characterization rather than to a reconstitution of the RTCB-Archease complex.

2) The authors should elaborate on what is shown in Figure 1D, as they probably refer to reaction kinetics (turnover numbers) and not half-lives. The unit 'mi' should read 'min'.

3) The color coding in Figure 2A is not explained in the caption. Therefore, it is difficult to realize

that the presented structure of Archease is indeed monomeric. The domain swap observed in the crystal structure of *P. horikoshii* Archease is well presented in Extended Data Figure 3B and could be omitted in Figure 2 for clarity.

4) Figure 3 should be merged with Extended Data Figure 4 as the results are identical. Furthermore, it is advisable to identify RTCB as a component of the cross-linked species by immunoblotting.

5) Please provide an additional short exposure image of the guanylylation assay provided in Figure 5C, since the two time points of WT can hardly be differentiated.

6) Please label the shifted His227 of activated RTCB in Figure 6B (right panel) for better clarity. Consider rephrasing the term 'post-activation complex' to 'activated RTCB' complex. Revisit panel D, as the difference of the red area between the blue and grey molecules is not evident.

7) In the very nice overview presented in Figure 7 the authors may consider indicating missing structures by a transparent shading. In this way, the structures determined in the present study will get more attention.

8) The title of Extended Data Fig. 2 may be changed to 'Effect of Archease and nucleotide on RTCB-catalyzed RNA ligation.'

9) Data presented in Extended Data Fig. 5 could in part be moved to main text Figure 2, whereas the domain swap found in other Archease structures could be highlighted in an Extended Data Figure.

10) The following statements should be referenced:

- line 25: 'Among the various types...comprised of tRNA ligases.'
- line 26: 'These enzymes are... tRNA splicing machinery.'
- line 69: 'Isolated and pre-activated...upon addition of Archease.'

11) The authors define three Mn²⁺ ion positions and refer to them as A, B, and C. Readability will improve if these were mentioned consistently as e.g. 'Mn²⁺ ion in position A' instead of 'Mn²⁺ ion (A)', as the '(A)' can be easily confused with a reference to a figure panel.

12) The authors use the terms Mn²⁺ ion, Mn²⁺ cation, and manganese ion interchangeably throughout the manuscript. To improve readability, one expression may be considered throughout and only changed when appropriate.

13) line 54: 'Crystal structures of RTCB...archaeon *Pyrococcus horikoshii*17,18,21,22.' This sentence is difficult to read and should be rephrased. Are the folds of canonical ligases conserved? What are the structural differences between folds of canonical ATP-dependent ligases and RTCBs?

14) line 106 ff: The authors should embed the paragraph entitled 'Structure of human Archease reveals a compact monomeric fold' at a later stage in the manuscript and first present their data on the ternary pre-activation complex.

15) line 119: 'Residues 22-47 including...Extended Data Fig. 3A).' It is unclear to which structure of human Archease the authors refer here since Figure 2C depicts the level of sequence conservation of monomeric Archease and Extended Data Figure 3A shows the domain-swapped structure of an archaeal Archease. Please correct the sentence and provide a PDB ID.

16) line 208: 'Moreover, Y92 undergoes one of the largest sidechain rotations in RTCB upon complex formation,...'. Please show the movement of Y92 in a figure, like H227 of RTCB in Figure 6B.

17) line 239: 'In the RTCB active site, five conserved residues bind either Mn²⁺ (A) or Mn²⁺ (B) (Fig. 4C).' The identities of the conserved residues should be mentioned in the text.

18) line 331-334. The authors should refer to Figure 7 after the sentence.

19) line 455 and following: protein concentrations should be reported consistently with one decimal place and proper unit (mg/mL, mg/ml, or mg ml⁻¹).

20) line 484: the word 'Figures' should not be capitalized.

21) line 499: The distributor of α 32P-GTP should be mentioned.

22) In some of their sentences it came to my attention that the authors mix past and present tense, e.g. line 234 'In the former case, there was no detectable guanylylation of RTCB (Fig. 5C, lane 7) while the latter displays diminished guanylylation (Fig. 5C, lane 8).' (plus lines 313-314, 326-329). Please double check.

Reviewer #3 (Remarks to the Author):

The manuscript by Gerber et al. described the structural and mechanistic insights into the activation of the human RNA ligase RTCB by Archease. The authors solved four crystal structures for the human RTCT-Archease complex. They revealed the structural snapshots along the reaction cycle of the tRNA ligase complex. This work is crucial for understanding the RNA ligase during tRNA splicing, unfolded protein response, and repair of RNA. However, the manuscript is unsuitable for publication in Nat Commun until the concerns below are addressed.

1, The four crystal structures were over-refined. Table 1, the outer shell I/sigma values (PDB: 8BTX, 8ODP, 8ODO, and 8BTT) ranged from 0.9 to 1.5, which were unacceptable for refinements. Also, the Rmerge values were too high to be acceptable. The authors should re-refine the four structures and decrease the resolutions to some reasonable ranges, e.g., I/sigma > 2.0 for the outer shell.

2, Figure 2, as human Archease crystal structure was reported in the paper (Duan et al., IJBCB, 2020), it is necessary to compare the difference between the structures (8BTX vs 5YZ1). Also, this figure is suitable to be present as the supplementary material.

3, Figure 4, for RTCB-Archease complex structure, the authors must show the omit density maps for GMP and GMPCC. Also, how to identify the metal was Mn²⁺ instead of Mg²⁺ in the crystal structure (figure 4C)? ICP (inductively coupled plasma) or X-ray scattering results will be helpful.

NCOMMS-23-25571-T

“Structural and mechanistic insights into activation of the human RNA ligase RTCB by Archease”

Response to Reviewers' Comments

Reviewer #1 (Remarks to the Author):

In this work, Gerber and colleagues provide the mechanistic basis for the activation of the human RTCB-Archease complex, by imaging the complex in its pre- and post-activation states. Their work is complemented by biochemical work, where the authors also reconstitute the ligation reaction in vitro.

RTCB-type ligases are present in all domains of life and, like canonical Trl1-type ligases, they ligate RNA ends carrying a 2',3'-cyclic phosphate and a 5'-OH. In tRNAs these ends are generated by TSEN complex. In addition, the tRNA ligase complex also ligates XBP1 mRNA, which is a crucial component of the unfolded protein response (UPR).

Human RTCB is an integrated part of the pentameric human tRNA ligase complex, which also harbors of DDX1, CGI-99, FAM98B and ASHWIN. The ligation reaction by the tRNA ligase complex is known to require GTP-dependent activation of RTCB. During the reaction RTCB undergoes guanylylation, which depends on the on an activation factor, named Archease.

First, the authors reconstitute the ligation reaction of XBP1 mRNA fragments after cleavage by IRE1, using purified human RTCB and Archease proteins. They further show that the multi-turnover reaction as well as the interaction between the two protein partner depend on the presence of GTP and Mn. Next, they determine several high-resolution crystal structures of human Archease, RTCB and the RTCB-Archease complex in its pre- and post-activation states. They use these structures to compare the conformation of the active site residues and validate their structural findings with structure-guided mutants.

The work closes several important knowledge gaps by structurally resolving important intermediates of the reaction. In conclusion, I would like to congratulate the authors for their work and I support publication of the manuscript in Nature Communications after resolving a few minor issues.

Obviously, there is one major remaining issue that is not touched experimentally at all - how is the mechanism incorporated in the fully assembled human tRNA ligase complex and does it work slightly different for tRNA substrates. However, the presented work provides a solid mechanistic basis to precisely define those next challenges, which definitely need to be addressed in the future.

We thank the reviewer for the positive feedback and the constructive comments. We agree that the tRNA ligase/RTCB mechanism in the context of the holo-ligase complex remains amongst the next important challenges. We anticipate that our results on the Archease-dependent activation mechanism of RTCB including the mode of interaction will be fundamentally the same in the presence of the additional subunits. However, the

fully assembled tRNA ligase complex might exhibit additional layers of regulation and fine-tuning during the individual steps of the reaction cycle.

Minor issues

Please define the RTCB abbreviation for the non-expert reader

We have added the full name in the text (line 38).

I feel that the last sentence of the abstract is an overstatement “Taken together, we present structural snapshots along the reaction cycle of the tRNA ligase complex.” Please rephrase accordingly – the work does not provide any experimental evidence that the mechanisms works identically within the ligase complex – even if one can assume it is the case.

We have changed “tRNA ligase complex” to “human tRNA ligase” to remove the notion of holo-complex structures.

“in vitro” should be italic

According to the Nature Communications style, we have kept “in vitro” and similar Latin-derived phrases non-italic.

Page 8 – line 143 – “Based on the results of the crosslinking experiments,...”. I am slightly confused, if the crystallization was performed using cross-linked complexes – from the M&M section, it seems that the proteins were just used at high concentrations and no cross-linker was used. Please rephrase the text to avoid any confusion. Please mention if any reconstitution experiments were performed and if the complex did not withstand any gel filtration approaches (which is not uncommon for dynamic complexes).

We thank the reviewer for pointing out possible misinterpretations. All crystallization trials in this study were performed without prior crosslinking. We have rephrased the first sentence of this paragraph accordingly to “Since crosslinking experiments suggested a stabilization of the complex in the presence of its cofactors, we attempted crystallization...” (page 8, line 145ff). The respective crystallization methods are described with all components and their concentrations in the Methods section as well as the deposited PDB entries.

In a previous study by the Jinek Lab (Kroupova et al., eLife, 2021), the authors show that Archease elutes separately from the ligase core complex when analyzed by size-exclusion chromatography. We have not repeated a gel filtration experiment ourselves. However, we have attempted to characterize the RTCB-Archease interaction using various in vitro binding assays, including FRET, fluorescence anisotropy, intrinsic Trp fluorescence, bio-layer interferometry and thermophoresis. None of these approaches allowed us to monitor complex formation, presumably, due to the dynamic nature of the complex.

In Extended Data Fig. 1, the authors show a SDS-PAGE gel of the purified proteins – it would be good to also show the Absorbance profile to show that both proteins did not purify with any bound RNA contaminants. Could such bound RNA molecules be the reason for problems to reconstitute the complex in vitro? Was the sample ever applied to a Heparin column to remove bound Nucleic acids?

As the reviewer correctly points out, it is important to remove bound RNA (or DNA) contaminants when purifying RTCB (as with most RNA-binding proteins). To this end, we have used buffers with increased ionic strengths (i.e. 300 mM NaCl) during lysis and the initial Ni affinity chromatography step. Instead of the suggested heparin column, we have used anion exchange chromatography with a strong anion exchange matrix (Q), which yielded separation of nucleic acid contaminants as indicated by a secondary elution peak with higher absorbance at 260 nm. We have measured absorbance spectra and calculated the 260/280 ratio for both proteins. The respective values are 0.55 for RTCB and 0.56 for Archease indicating absence of RNA contaminants. The values for the ratios are stated in the Methods section and the full spectra are documented in the source data.

The authors conclude that all observed sites are Mn sites – were any efforts undertaken to measure and calculate anomalous difference Fourier maps at Mn-specific wavelengths? If not, please make a comment about the remaining uncertainty of the identity of the atom. Mg could have been co-purified during the purifications. Same applies to the phosphate atoms of the potential pyrophosphate in the guanylated post-activation complex.

Sebastian Glatt

The reviewer raises an important point about the identity of the observed metal sites in our complex structures. Here, we would like to explain why manganese is the most probable identity of the metal sites and how we derived this conclusion from our data. Unfortunately, we have not collected diffraction data at or close to the K edge of manganese (i.e. 1.89 Å). Thus, we could not calculate anomalous difference Fourier maps as asked by the reviewer. Instead, we have recorded X-ray fluorescence (XRF) spectra of the complex, which confirm the presence of Mn²⁺ ions in the complex structures. We have included this data as Source Data (see Figure below).

However, due to the limited detection range available at the ESRF beamline, the XRF spectra do not contain data at the K edge of magnesium. Thus, the XRF spectra alone do indicate the presence of Mn^{2+} ions in the crystal, but cannot exclude the presence of Mg^{2+} directly. To prevent (or minimize) the carry-over of Mg^{2+} during the purification process, we have (as described in Methods) treated the proteins during purification with an excess of the chelating agent EDTA followed by dialysis.

Furthermore, our metal reconstitution assays (Fig. 5c & 5d) showed that there is neither RTCB guanylylation nor ligase activity without the addition of metal ions. These results indicate Mg^{2+} -free protein preparations since the addition of Mg^{2+} yielded some activity (albeit much reduced compared to Mn^{2+} addition). We would like to emphasize that the same purification batches were used for both, crystallization and biochemical analysis.

A close look at the XRF data, however, shows a secondary peak for zinc. In consequence, we cannot rule out partial occupancy of the metal sites by Zn^{2+} ions. Nonetheless, our metal reconstitution experiments provide evidence for the strongly inhibiting property of Zn^{2+} on RTCB activity. We surmise that only a minor fraction of the metal sites might be occupied by Zn^{2+} while the majority is occupied by the added metal ion, which is Mn^{2+} in case of crystallization. As discussed in our manuscript, zinc as a softer Lewis acid probably binds stronger than manganese in positions A and B due to the conserved cysteine and histidine residues in the active site. This specific ligand environment of metal-binding sites A and B further argues against co-purification of Mg^{2+} . As a slightly larger ion and a softer Lewis acid, Mn^{2+} has a much higher affinity for coordination environments containing more than one non-oxygen ligand, i.e. nitrogen or sulfur [Tari et al., *Nature Struct Biol*, 1997; Bock et al., *JACS*, 1999].

In summary, we argue that the combination of our rigorous EDTA-treatment, positive confirmation of manganese through XRF spectrometry, our biochemical reconstitution upon metal addition as well as the environment of the metal-sites renders Mn^{2+} the most probable metal ions in our complex crystal structures.

In our view, there is an additional important aspect regarding this issue: What is the identity of the metal ions *in vivo*? The question cannot be answered by our *in vitro* reconstitution approach but we have acknowledged the possibility of mixed metal ion occupancy in the discussion.

Co-purification of phosphate is in our opinion rather unlikely. None of the buffers

contained phosphate. Neither our nucleotide-free structure nor any previously published RTCB structures contain phosphate at this site implying an interaction not strong enough to withstand the regular chromatographic purification process. Most importantly, we know that phosphate is being produced during guanylation. Thus, phosphate is present in the crystallization mix as product of the activation reaction in any case.

Reviewer #2 (Remarks to the Author):

The article ‘Structural and mechanistic insights into activation of the human RNA ligase RTCB by Archease’ by Gerber et al. reports on various structural snapshots of human RTCB, the active component of the human tRNA ligase complex, during its catalytic cycle. Besides crystal structures of human Archease and RTCB in the apo state, the authors captured a transient ternary complex of RTCB, Archease, and the GTP analog GMPCPP and solved its crystal structure. Furthermore, they present a crystal structure of activated, guanylated RTCB in complex with Archease. The authors corroborate their structural data with functional analyses that shed light on the metal ion dependency of RTCB guanylation catalyzed by Archease and the Archease dependence of RTCB-catalyzed RNA ligation. The clear novelty of the work by Gerber et al. lies in, first, capturing transient chaperoning and activating states of the human transfer RNA ligase RTCB, and second, in determining structural snapshots of these states that significantly augment our understanding of RTCB-catalyzed RNA ligation.

In my opinion, the manuscript will benefit from a restructured presentation of the data focusing on RTCB rather than Archease. Therefore, the authors should consider introducing and describing their crystal structure of the ternary RTCB-Archease-GMPCPP complex in the pre-activated state first, then presenting the activated state of RTCB, and eventually discuss the structures of the individual components including the structural dynamics that are seen upon complex formation—in a similar order as written in their abstract.

We thank the reviewer for the overall positive evaluation of our work and the constructive feedback on the structure of our manuscript. We are responding to the reviewer’s suggestions in the following point-to-point reply.

Major:

The authors should show an overview of the interaction surface(s) between RTCB and Archease in the main text and discuss interacting residues in a separate figure, which may be referred to in line 151 after ‘...with an area of about 2,200 Å².’ Do the crystal structures explain the transient character of the complex? As the authors provide evidence that fluorescently labelled Archease chemically cross-links to recombinant RTCB, fluorescence anisotropy measurements may be considered for determination of dissociation constants (K_d) of the ternary complex to further define its transient character.

We have added a visual overview of the interaction surfaces in Supplementary Data Figure 4a as well as the suggested reference in line 151 (now line 155). In addition, we have compiled a list of all protein-protein interactions with the corresponding residues according to a PISA analysis in the new Supplementary Table 1.

As stated above in our response to reviewer #1, we have attempted to characterize the RTCB-Archease interaction using various in vitro binding assays, including FRET,

fluorescence anisotropy, bio-layer interferometry, intrinsic Trp fluorescence and thermophoresis. None of these approaches allowed us to monitor complex formation. Other studies came to similar conclusions. The aforementioned study by Kroupova et al. (eLife, 2021) showed a separate elution peak of Archease from the RTCB-containing core complex during gel-filtration. In addition, in a paper by the Raines Lab (Desai et al., RNA Journal, 2015) on the co-evolution of RTCB and Archease, the direct binding between both proteins from *Pyrococcus horikoshii* could not be monitored. The authors concluded that they “were unable to detect a physical interaction between the two proteins, suggesting that they interact only transiently”.

We find it difficult, if not impossible, to speculate on the dynamics of the complex based on our structure and biochemical assays. Frankly, we are puzzled by the disconnect between the presented data, which are all in line with productive formation of the RTCB-Archease complex over a large concentration range, and the difficulties with determining a dissociation constant. In our view, a thorough, quantitative characterization of the RTCB-Archease interaction is needed in the future, but beyond the scope of the current manuscript.

Please revisit items that are presented in main Figures and Extended Data Figures (as pointed out below). The paragraph entitled ‘Crystal structures of the RTCB-Archease complex in two distinct nucleotide-bound states’ should be split in two parts (between lines 185 and 186) to logically separate the descriptions of the pre-activation and the activated state of RTCB. The current paragraph reads as if the authors accidentally obtained the activated state of RTCB. Please substantiate the sentence ‘The position of the covalent bond at the α -phosphate was inverted due to the phosphoryl transfer reaction,...’ (lines 192-193).

We have taken the reviewer’s suggestion to split the respective paragraphs by introducing additional sub-headings. In addition, we have revised the introductory sentence for the post-activation complex structure to exclude the notion of having obtained this state “accidentally”. The paragraph now starts with: “To obtain a structure of the RTCB-Archease complex in the post-activation state (i.e. with the activated RTCB-GMP intermediate), we co-crystallized RTCB with Archease in the presence of the nucleotide co-factor GTP, and Mn^{2+} ions.” (line 192 ff).

We agree with the reviewer that ‘the inverted covalent bond at the α -phosphate’ requires additional substantiation. We have added a hint for the reader to “compare left to right panel in Fig. 4c” and extracted the important changes to a new supplementary figure (Supplementary Fig. 5g).

Minor:

Consider changing the title of Figure 1 as the data refers to a functional characterization rather than to a reconstitution of the RTCB-Archease complex.

We have changed the title to “In vitro splicing assay of the RTCB/Archease RNA ligase.”.

The authors should elaborate on what is shown in Figure 1D, as they probably refer to reaction kinetics (turnover numbers) and not half-lives. The unit ‘mi’ should read ‘min’.

The simple experimental setup does not allow direct calculation of turnover numbers. However, to use a unit that is more commonly associated with reaction kinetics, we provide the apparent rate constant in min^{-1} .

We have corrected the unit to 'min'.

*The color coding in Figure 2A is not explained in the caption. Therefore, it is difficult to realize that the presented structure of Archease is indeed monomeric. The domain swap observed in the crystal structure of *P. horikoshii* Archease is well presented in Extended Data Figure 3B and could be omitted in Figure 2 for clarity.*

We thank the reviewer for pointing out the lacking explanation of the color coding. We have added the explanation of the color coding of Figure 2a in the figure legend. The initial version of the figure was missing further important information. Figure 2a is a superposition of the dimeric **human** Archease structure (PDB ID 5YZ1) and our monomeric structure. For clarity, this information was added in the text and figure legend. The superposition with *P. horikoshii* Archease in Supplementary Figure 3b remains unaffected by these changes, as we are comparing our monomer structure to both previously solved dimeric Archease crystal structures.

Figure 3 should be merged with Extended Data Figure 4 as the results are identical. Furthermore, it is advisable to identify RTCB as a component of the cross-linked species by immunoblotting.

We agree with the reviewer's comment that the respective figures were (almost) identical in the initial version of the manuscript. The purpose of the supplementary data (originally Extended Data Figure 4) was to identify Archease as a component of the cross-linked species. Considering the reviewer's suggestion to identify RTCB in the cross-linked species as well, we have now included a similar experiment with both proteins being fluorescently labeled (new Fig. 3b). Due to the lack of specific antibodies against RTCB (or Archease), we have selected this experimental strategy over immunoblotting. The merge (Fig. 3b, left panel) with RTCB and Archease each labeled with a different fluorophore confirms the presence of both proteins in the cross-linked species. In consequence, we have omitted the old Extended Data Figure 4, which is redundant at this point.

Please provide an additional short exposure image of the guanylylation assay provided in Figure 5C, since the two time points of WT can hardly be differentiated.

We thank the reviewer for pointing this out. Since we have recorded the initial image as exposure series, we have added an additional image with shorter exposure time to Figure 5e.

Please label the shifted His227 of activated RTCB in Figure 6B (right panel) for better clarity. Consider rephrasing the term 'post-activation complex' to 'activated RTCB' complex. Revisit panel D, as the difference of the red area between the blue and grey molecules is not evident.

We have labeled the shifted His227 residue in the respective color of the sidechain (teal) for better clarity.

We have considered changing the naming of the complex. However, we have decided to stick with the term 'post-activation complex' because we (and others) refer to guanylated RTCB as 'activated' also in the absence of Archease. Thus, we have decided to use the more unique term to avoid any misunderstanding.

We agree with the reviewer that the difference may be difficult to see in a still image. Thus, we have added a video of the morph between both conformations to illustrate the shift of the active site pocket as Supplementary Movie 1.

In the very nice overview presented in Figure 7 the authors may consider indicating missing structures by a transparent shading. In this way, the structures determined in the present study will get more attention.

We thank the reviewer for the input on our depiction of the reaction cycle, which we consider an important graphical summary of our manuscript as well as the current state of knowledge. Thus, we have included the reviewer's suggestion and have changed the color of the missing structures in the RTCB ligation cycle (i.e. the RTCB-substrate and RTCB-product complex) to grey. We have added one sentence to the legend as explanation.

The title of Extended Data Fig. 2 may be changed to 'Effect of Archease and nucleotide on RTCB-catalyzed RNA ligation.'

We have changed the figure title accordingly.

Data presented in Extended Data Fig. 5 could in part be moved to main text Figure 2, whereas the domain swap found in other Archease structures could be highlighted in an Extended Data Figure.

We acknowledge the reviewer's suggestion in light of the proposed changed order of the presented crystal structures (see our response to this point). However, as we prefer to introduce the complex-forming monomeric Archease structure first, we have decided to keep main text Fig. 2 and the supplementary data about Archease in the complex separately.

The following statements should be referenced:

- line 25: 'Among the various types...comprised of tRNA ligases.'
- line 26: 'These enzymes are... tRNA splicing machinery.'
- line 69: 'Isolated and pre-activated...upon addition of Archease.'

We have added combined references after the statements in lines 25 and 26 (now 26 and 27). The sentence 'Isolated and pre-activated...upon addition of Archease.' was initially referenced together with the following sentence, which we have changed in the revised version accordingly.

The authors define three Mn²⁺ ion positions and refer to them as A, B, and C. Readability will improve if these were mentioned consistently as e.g. 'Mn²⁺ ion in position A' instead of 'Mn²⁺ ion (A)', as the '(A)' can be easily confused with a reference to a figure panel.

The currently used descriptors of the metal ion sites follows the naming introduced by Kroupova et al. (eLife, 2021), which reports the first structure of human RTCB.

Consequentially, we have used the same naming and expanded it by the newly identified site 'C'. We have added the following short remark to avoid any confusion with figure panels: "These Mn²⁺ ions occupy three distinct positions A, B and C (hereinafter indicated in parentheses)." (page 9, lines 172 and 173). In addition, we have changed panel labels (according to Nature style) to lower-case letters, which prevents any confusion.

The authors use the terms Mn²⁺ ion, Mn²⁺ cation, and manganese ion interchangeably throughout the manuscript. To improve readability, one expression may be considered throughout and only changed when appropriate.

We thank the reviewer for the great attention to detail. We have indeed used the terms interchangeably. We have changed these instances to 'Mn²⁺ ion' throughout the text. 'Manganese' is now only being used when referring to the element but not together with 'ion'. We have kept the term 'cation' only when used without the respective metal ('M²⁺').

line 54: 'Crystal structures of RTCB...archaeon Pyrococcus horikoshii17,18,21,22.' This sentence is difficult to read and should be rephrased. Are the folds of canonical ligases conserved? What are the structural differences between folds of canonical ATP-dependent ligases and RTCBs?

We have re-phrased the sentence accordingly. The adenylyltransferase fold of canonical ATP-dependent ligases is conserved. It is difficult to discuss the differences between both types of tRNA ligases as do they do not share common origin. In fact, the paper by Okada et al. [Crystal structure of an RtcB homolog protein (PH1602-extein protein) from Pyrococcus horikoshii reveals a novel fold] reported the RTCB structure as entirely novel fold compared to all PDB structures (even prior to its functional characterization as RNA ligase). The cited references discuss the differences between RTCB-type and adenylyltransferase-type RNA ligases in more detail.

line 106 ff: The authors should embed the paragraph entitled 'Structure of human Archease reveals a compact monomeric fold' at a later stage in the manuscript and first present their data on the ternary pre-activation complex.

We thank the reviewer for constructive suggestions on the structure of our manuscript. We have indeed considered the suggested order during initial manuscript preparation. However, there is one part that—in our opinion—logically demands the introduction of the Archease monomer structure, which is the description of conformational changes within RTCB and Archease upon complex formation. Since the existing dimeric Archease structure (Duan et al, PDB 5YZI) would sterically not allow 1:1 activation complex formation, we would first need to point out this discrepancy, only to correct it later when introducing our Archease structure. The situation is different for RTCB, where the existing structure is not at odds with activation complex formation. Therefore, the nucleotide-free structure of RTCB is presented last. We agree with the reviewer that the complex structures are the most important findings presented here, which is the reason for the changed order in the Abstract.

line 119: 'Residues 22-47 including...Extended Data Fig. 3A).' It is unclear to which structure of human Archease the authors refer here since Figure 2C depicts the level of sequence conservation of monomeric Archease and Extended Data Figure 3A shows the domain-

swapped structure of an archaeal Archease. Please correct the sentence and provide a PDB ID.

We thank the reviewer for pointing out that the initial referencing of Figures 2 and S3 contained some errors. We have checked all references to the respective figures and changed accordingly where needed. The superposition with the human Archease dimer (Fig. 2a) or *P. horikoshii* Archease (Supplementary Figure 3b) was not clear (as indicated by all three reviewers) due to the missing PDB ID for the human Archease dimer structure (PDB 5YZ1). We have added the PDB ID accordingly in the text and legend.

line 208: 'Moreover, Y92 undergoes one of the largest sidechain rotations in RTCB upon complex formation, ...'. Please show the movement of Y92 in a figure, like H227 of RTCB in Figure 6B.

We have added an additional panel as Supplementary Figure 5j to show the movement of Y92 between the RTCB-Archease complex and the nucleotide free RTCB.

line 239: 'In the RTCB active site, five conserved residues bind either Mn²⁺ (A) or Mn²⁺ (B) (Fig. 4C).' The identities of the conserved residues should be mentioned in the text.

We now list the residues. The sentence reads as follows: 'In the RTCB active site, five conserved residues bind either Mn²⁺ (A) or Mn²⁺ (B), which are D119, C122, H227, H259 and H353'.

line 331-334. The authors should refer to Figure 7 after the sentence.

We have added a reference to Figure 7 accordingly.

line 455 and following: protein concentrations should be reported consistently with one decimal place and proper unit (mg/mL, mg/ml, or mg ml⁻¹).

We have changed all protein concentrations to one decimal place. Concentrations are now given in mg/mL throughout.

line 484: the word 'Figures' should not be capitalized.

We have changed it to lowercase 'figure' accordingly.

line 499: The distributor of α 32P-GTP should be mentioned.

We have added the distributor (Hartmann Analytic) in the Methods section.

In some of their sentences it came to my attention that the authors mix past and present tense, e.g. line 234 'In the former case, there was no detectable guanylylation of RTCB (Fig. 5C, lane 7) while the latter displays diminished guanylylation (Fig. 5C, lane 8).' (plus lines 313-314, 326-329). Please double check.

We have made those changes accordingly.

Reviewer #3 (Remarks to the Author):

The manuscript by Gerber et al. described the structural and mechanistic insights into the activation of the human RNA ligase RTCB by Archease. The authors solved four crystal structures for the human RTCB-Archease complex. They revealed the structural snapshots along the reaction cycle of the tRNA ligase complex. This work is crucial for understanding the RNA ligase during tRNA splicing, unfolded protein response, and repair of RNA. However, the manuscript is unsuitable for publication in Nat Commun until the concerns below are addressed.

We thank the reviewer for considering our work 'crucial for understanding the tRNA ligase' in its various cellular processes. We have responded to the critical points regarding crystallographic data processing and visualization of our structural models below.

The four crystal structures were over-refined. Table 1, the outer shell I/sigma values (PDB: 8BTX, 8ODP, 8ODO, and 8BTT) ranged from 0.9 to 1.5, which were unacceptable for refinements. Also, the Rmerge values were too high to be acceptable. The authors should re-refine the four structures and decrease the resolutions to some reasonable ranges, e.g., I/sigma > 2.0 for the outer shell.

We are appreciative for initiating the discussion on the important subject of resolution limits. However, we are convinced that we chose appropriate resolution cutoffs for all four crystal structures. For data reduction, scaling and the determination of the maximum resolution, we used the program AIMLESS, which is included in the CCP4 package. AIMLESS is considered the standard program for data reduction and scaling in protein X-ray crystallography. As described by Evans & Murshudov (Acta Cryst, 2013), the program AIMLESS considers several parameters including anisotropy for quality assessment and for the estimation of the maximum resolution. Moreover, the half-data set correlation $CC_{1/2}$ as proposed by Karplus & Diederichs (Science, 2012) and supported by Evans (Science 2012) is now regarded as the main quality indicator for crystallographic data sets. Therefore, we applied the exact overall resolution estimate based on $CC_{1/2}$ given by AIMLESS as high-resolution limit for all four data sets.

We have added a short description of our choice of resolution cutoffs in the Methods section and have referenced both above-mentioned, fundamental papers on the topic. Thus, we think that the reviewer's comment promoted a more thorough and clearly defined description of our data processing procedures.

Figure 2, as human Archease crystal structure was reported in the paper (Duan et al., IJBCB, 2020), it is necessary to compare the difference between the structures (8BTX vs 5YZ1). Also, this figure is suitable to be present as the supplementary material.

We also thank reviewer #3 for pointing out the lacking description with regards to Figure 2. As stated above in our responses to reviewers #1 and #2, the initial referencing of Figures 2 and S3 contained some errors. We have checked all references to the respective figures and changed accordingly where needed. The requested superposition with the human Archease dimer (PDB 5YZ1) was included in Fig. 2a but unfortunately not labeled as such. In Supplementary Figure 3b we show the superposition of our structure with the *P. horikoshii* Archease dimer structure. We have added all PDB IDs accordingly in the text and legends.

Figure 4, for RTCB-Archease complex structure, the authors must show the omit density maps for GMP and GMPCC. Also, how to identify the metal was Mn²⁺ instead of Mg²⁺ in

the crystal structure (figure 4C)? ICP (inductively coupled plasma) or X-ray scattering results will be helpful.

We have added the omit density map for GMPCPP in Supplementary Figure 4f.

[Please note that we have duplicated large parts of the following reply from our answer above to almost the same question by reviewer #1.] The reviewer raises an important point about the identity of the observed metal sites in our complex structures. Here, we would like to explain why manganese is the most probable identity of the metal sites and how we derived this conclusion from our data. We have neither collected diffraction data at or close to the K edge of manganese (i.e. 1.89 Å) nor used Inductively coupled plasma mass spectrometry. Instead, we have recorded X-ray fluorescence (XRF) spectra of the complex, which confirm the presence of Mn^{2+} ions in the complex structures. We have included this data as Source Data (see Figure below).

However, due to the limited detection range available at the ESRF beamline, the XRF spectra do not contain data at the K edge of magnesium. Thus, the XRF spectra alone do indicate the presence of Mn^{2+} ions in the crystal, but cannot exclude the presence of Mg^{2+} directly. To prevent (or minimize) the carry-over of Mg^{2+} during the purification process, we have (as described in Methods) treated the proteins during purification with an excess of the chelating agent EDTA followed by dialysis.

Furthermore, our metal reconstitution assays (Fig. 5c & 5d) showed that there is neither RTCB guanylation nor ligase activity without the addition of metal ions. These results indicate Mg^{2+} -free protein preparations since the addition of Mg^{2+} yielded some activity (albeit much reduced compared to Mn^{2+} addition). We would like to emphasize that the same purification batches were used for both, crystallization and biochemical analysis.

A close look at the XRF data, however, shows a secondary peak for zinc. In consequence, we cannot rule out partial occupancy of the metal sites by Zn^{2+} ions. Nonetheless, our metal reconstitution experiments provide evidence for the strongly inhibiting property of Zn^{2+} on RTCB activity. We surmise that only a minor fraction of the metal sites might be occupied by Zn^{2+} while the majority is occupied by the added metal ion, which is Mn^{2+} in case of crystallization. As discussed in our manuscript, zinc as a softer Lewis acid probably binds stronger than manganese in positions A and B due to the conserved cysteine and histidine residues in the active site. This specific ligand environment of metal-binding sites A and B further argues against co-purification of Mg^{2+} . As a slightly larger ion and a softer Lewis acid, Mn^{2+} has a much higher affinity for

coordination environments containing more than one non-oxygen ligand, i.e. nitrogen or sulfur [Tari et al., *Nature Struct Biol*, 1997; Bock et al., *JACS*, 1999].

In summary, we argue that the combination of our rigorous EDTA-treatment, positive confirmation of manganese through XRF spectrometry, our biochemical reconstitution upon metal addition as well as the environment of the metal-sites renders Mn^{2+} the most probable metal ions in our complex crystal structures.

Reviewer #1 (Remarks to the Author):

In this work, Gerber and colleagues provide the mechanistic basis for the activation of the human RTCB-Archease complex, by imaging the complex in its pre- and post-activation states. Their work is complemented by biochemical work, where the authors also reconstitute the ligation reaction in vitro.

I have re-evaluated the revised version of the manuscript and detailed point-by-point response provided by the authors. I appreciate the inclusion of the additional data in the source data section that addresses the minor issues raised during the first round of review – in particular, the provided absorbance spectra directly answer my question concerning possible nucleic acid contaminations. The response of the authors concerning the identity of the identified atoms in the active site is less satisfying - it is indeed a pity that the respective anomalous datasets (on the zinc and Mn edge) have not been collected. In particular, as the X-ray fluorescence spectroscopy analyses have been performed and the accessible peaks have been detected. Considering the complementary biochemical analyses, I can follow the conclusions drawn by the authors. However, I would still suggest to mention the detection of zinc by XRF in the manuscript – similar to the mentioning in the response letter. I feel that in the moment the sentence on page 8 - "The presence of Mn²⁺ ions in the complex structures was confirmed by X-ray fluorescence spectroscopy (see Source Data)." - makes the opposite impression that nothing else than Mn has been found in the analyses. Otherwise, I support publication of the manuscript in Nature Communications.

Sebastian Glatt

Reviewer #2 (Remarks to the Author):

My comments from the initial round of review have been sufficiently addressed by the authors and I support publishing this work in Nature Communications.

Reviewer #3 (Remarks to the Author):

The manuscript by Gerber et al. described the structural and mechanistic insights into the activation of the human RNA ligase RTCB by Archease. I appreciate the authors' efforts to improve the manuscript quality in the revised version. However, there are still some concerns the authors should address before it can be published.

The original question is:

The four crystal structures were over-refined. Table 1, the outer shell I/σ values (PDB:8BTX, 8ODP, 8ODO, and 8BTT) ranged from 0.9 to 1.5, which were unacceptable for refinements. Also, the Rmerge values were too high to be acceptable. The authors should re-refine the four structures and decrease the resolutions to some reasonable ranges, e.g., $I/\sigma > 2.0$ for the outer shell.

The authors' reply:

We are appreciative for initiating the discussion on the important subject of resolution limits. However, we are convinced that we chose appropriate resolution cutoffs for all four crystal structures. For data reduction, scaling and the determination of the maximum resolution, we used the program AIMLESS, which is included in the CCP4 package. AIMLESS is considered the standard program for data reduction and scaling in protein X-ray crystallography. As described by Evans & Murshudov (Acta Cryst, 2013), the program AIMLESS considers several parameters including anisotropy for quality assessment and for the estimation of the maximum resolution. Moreover, the half-data set correlation CC1/2 as proposed by Karplus & Diederichs (Science, 2012) and supported by Evans (Science 2012) is now regarded as the main quality indicator for crystallographic data sets. Therefore, we applied the exact overall resolution estimate based on CC1/2 given by AIMLESS as high-resolution limit for all four data sets. We have added a short description of our choice of resolution cutoffs in the Methods section and have referenced both above-mentioned, fundamental papers on the topic. Thus, we think that the reviewer's comment promoted a more thorough and clearly defined description of our data processing procedures.

---The AIMLESS program is OK for structure analysis, scaling, and determination. The half-data set correlation CC1/2, as proposed by Karplus & Diederichs (Science, 2012), is also a good quality indicator for high-resolution datasets. However, the outer shell I/ σ value is more reliable for indicating the crystal structure quality.

For example, the I/ σ value of the RTCB structure (PDB: 8BTT) is 0.9 at 2.6 Å, which is unacceptable. Also, 2.6 Å is not a high resolution for protein crystal structure, as discussed in the paper by Karplus & Diederichs (Science, 2012). Meanwhile, the outer shell CC1/2 value is 0.348, which is questionable. The same situation is applied to the Archease structure (PDB: 8BTX). Again, the four crystal structures were over-refined. I suggest the authors decrease the structure resolutions for PDBs 8BTT, 8ODO, and 8BTX.

NCOMMS-23-25571A

“Structural and mechanistic insights into activation of the human RNA ligase RTCB by Archease”

Response to Reviewers' Comments

Reviewer #1 (Remarks to the Author):

In this work, Gerber and colleagues provide the mechanistic basis for the activation of the human RTCB-Archease complex, by imaging the complex in its pre- and post-activation states. Their work is complemented by biochemical work, where the authors also reconstitute the ligation reaction in vitro.

I have re-evaluated the revised version of the manuscript and detailed point-by-point response provided by the authors. I appreciate the inclusion of the additional data in the source data section that addresses the minor issues raised during the first round of review – in particular, the provided absorbance spectra directly answer my question concerning possible nucleic acid contaminations.

The response of the authors concerning the identity of the identified atoms in the active site is less satisfying - it is indeed a pity that the respective anomalous datasets (on the zinc and Mn edge) have not been collected. In particular, as the X-ray fluorescence spectroscopy analyses have been performed and the accessible peaks have been detected. Considering the complementary biochemical analyses, I can follow the conclusions drawn by the authors. However, I would still suggest to mention the detection of zinc by XRF in the manuscript – similar to the mentioning in the response letter. I feel that in the moment the sentence on page 8 - “The presence of Mn²⁺ ions in the complex structures was confirmed by X-ray fluorescence spectroscopy (see Source Data).” - makes the opposite impression that nothing else than Mn has been found in the analyses.

Otherwise, I support publication of the manuscript in Nature Communications.

Sebastian Glatt

We thank the reviewer for the positive feedback and the constructive comments. We have incorporated the suggested change on page 8 of the manuscript to acknowledge the presence of a Zn peak in the XRF data. The sentence in question now reads: “Using X-ray fluorescence spectroscopy, we confirmed the presence of Mn²⁺ ions in the complex structures while lower amounts of Zn²⁺ ions were also detected (see Source Data).”.